# ETVs dictate hPSC differentiation by tuning biophysical properties

Natalia M. Ziojła [1,5], Magdalena Socha [1,5], M. Cecilia Guerra[2], Dorota Kizewska[1], Katarzyna Blaszczyk[1], Edyta Urbaniak[1], Sara Henry[1], Malgorzata Grabowska [1], Kathy K. Niakan [3], Aryeh Warmflash[2] & Malgorzata Borowiak [1,4] ✉

Stem cells maintain a dynamic dialog with their niche, integrating biochemical and biophysical cues to modulate cellular behavior. Yet, the transcriptional networks that regulate cellular biophysical properties remain poorly defined. Here, we leverage human pluripotent stem cells (hPSCs) and two morphogenesis models – gastruloids and pancreatic differentiation – to establish ETV transcription factors as critical regulators of biophysical parameters and lineage commitment. Genetic ablation of ETV1 or ETV1/ETV4/ETV5 in hPSCs enhances cell-cell and cell-ECM adhesion, leading to aberrant multilineage differentiation including disrupted germ-layer organization, ectoderm loss, and extraembryonic cell overgrowth in gastruloids. Furthermore, ETV1 loss abolishes pancreatic progenitor formation. Single-cell RNA sequencing and follow-up assays reveal dysregulated mechanotransduction via the PI3K/AKT signaling. Our findings highlight the importance of transcriptional control over cell biophysical properties and suggest that manipulating these properties may improve in vitro cell and tissue engineering strategies.

Stem cells engage in continuous dialog with their microenvironment through both biochemical and physical signals, allowing them to interpret, respond to, and even modify their surroundings. Mechanical forces emerge as critical physical cues that significantly influence stem cell behavior. In vivo, stem cell niche imposes constraints, and features such as cell shape, adhesion, reciprocal interactions with neighboring cells and the extracellular matrix (ECM), and cytoskeleton tension profoundly impact stem cell homeostasis and determine their fate[1–4]. Similarly, in vitro, biophysical cues shape the self-renewal and differentiation of human pluripotent stem cells (hPSCs), including human embryonic stem cells (hESCs), and induced pluripotent stem cells (iPSCs)[5,6]. Being epithelial cells, hPSCs rely on two primary adhesion modes: cadherins mediate cell–cell contacts[7] and integrins mediate adhesion to the ECM. Cadherin 1 (CDH1)-mediated cell–cell contacts maintain the hPSC colony core, while the peripheral cells bind to

extracellular fibers, primarily vitronectin, collagens, or laminins via focal adhesions[8–10]. Numerous integrin receptors on hPSCs enhance adhesion and self-renewal. For example, blocking vitronectin-binding integrins αvβ3 and αvβ5 significantly reduces hPSC adherence[11], and downregulation of integrin α6β1 decreases pluripotency marker expression like Nanog, Oct4, and Sox2[12]. Integrin receptors intracellularly bind to Paxillin (PXN), which connects to the actin cytoskeleton via talin and vinculin (VCL)[6,9]. Therefore, the actin cytoskeleton maintains cell shape and transmits signals between the cell and its surroundings.

Cell–cell and cell–matrix adhesion are essential for transmitting mechanical forces, which are converted into biochemical signals via mechanisms such as receptor accessibility changes and actin cytoskeleton remodeling[6,10,13,14]. Key signaling pathways critical for hPSC stemness and specification, including Hippo, TGF-β, WNT, EGF, and

[1]Institute of Molecular Biology and Biotechnology, Adam Mickiewicz University, Poznan, Poland. [2]Department of Biosciences, Rice University, Houston, TX, USA. [3]The Loke Centre for Trophoblast Research, Department of Physiology, Development and Neuroscience, University of Cambridge, Cambridge, UK. [4]McNair Medical Institute, Baylor College of Medicine, Houston, TX, USA. [5]These authors contributed equally: Natalia M. Ziojła, Magdalena Socha. ✉e-mail: malbor3@amu.edu.pl

PI3K/AKT, are mechanosensitive[15–18]. hPSCs sense and respond to adhesion strength, integrating mechanical cues with other signals to guide differentiation. Substrate rigidity, for instance, influences cell-ECM adhesion[19], proliferation[20], migration[21], cell–cell adhesion[22], and fate specification[23,24]. Mechanical interactions between cells and their surroundings are bidirectional: hPSCs remodel their microenvironment, by depositing proteins into ECM[25], degrading ECM components[26], exerting contractile forces[10], secreting signaling molecules, and adjusting receptor expression[27]. For instance, the changes in the actinomyosin cytoskeleton can deform ECM, creating tension and affecting how neighboring cells sense stiffness. Therefore, mechanical forces and adhesion cues reciprocally influence each other.

Mechanical cues transduced through adhesion sites and biochemical signals are crucial for processes such as cell segregation, directed migration, cell fate determination, and tissue boundary formation during embryogenesis[4,28]. Adhesion-mediated microenvironment–cell contacts regulate tissue shape and interactions, ultimately governing differentiation and morphogenesis[5]. For instance, cell–matrix and cell–cell interactions guide the spatial organization of germ layers during gastrulation[29–31] or segregation of pancreatic progenitors (PP) within the gut tube[32]. Disruptions in cell adhesion signaling or integrity of adhesion complexes can lead to developmental defects and pathological states. Similarly, in vitro, biophysical cues influence stem cell behaviors, such as mesenchymal stem cell differentiation, hematopoietic stem cell expansion, or neural lineage decisions[33–35].

While mechanobiology research predominantly focuses on external forces influencing cells (outside-in signaling), mechanosensing machinery, or mechanotransduction, the transcriptional mechanisms regulating how cells physically interact with their environment remain less understood. To understand the mechanisms controlling cell biophysical properties, we investigated cells undergoing extensive morphological changes during pancreatic development. Early pancreatic progenitors initially form an epithelial sheet, followed by the acquisition of endocrine or duct cell fate[36–38]. Endocrine progenitors (EP) marked by NEUROGENIN3 (Ngn3) expression[39] must leave the tightly packed epithelium to form endocrine islets dispersed throughout the pancreas[40,41]. To achieve this, changes in adhesion, cell size, cell communication, cell polarity, and migration are necessary. Our previous study identified four subtypes of Ngn3+ EPs in the murine pancreas. These subtypes reflect developmental progression, with EP1 activating the endocrine program, and EP4 expressing endocrine-specific genes and hormones[42]. Intriguingly, the EP2 subtype expresses genes associated with epithelial-mesenchymal transition (EMT), cell–cell, cell-ECM adhesion, and cell size alongside Ngn3, suggesting extensive remodeling[42]. Among the differentially expressed genes in EP2 compared to other EP subtypes, we identified the ETS transcription factor, ETV1 (ER81) as highly increased in EP2 suggesting its potential role in pancreatic progenitor remodeling.

The ETV1 transcription factors belong to the PEA3 subfamily of ETS factors, alongside ETV4 (PEA3) and ETV5 (ERM). All three members share a similar protein structure and recognize the same core DNA sequence (GGAA). However, microRNAs and post-translational modifications impose distinct functional roles[43–45]. ETVs regulate early development, organogenesis, and morphogenesis[46–48]. Conditional knockouts of Etv disrupt anterior-posterior patterning in the limb bud, branching morphogenesis of the lacrimal gland, and lens development controlled by FGF signaling[49–51]. In many cases Etv4 /Etv5 and Etv1 control other developmental processes[52,53]. In developing lungs, salivary glands, and kidneys, Etv4 and Etv5, present in the epithelium, regulate branching, while Etv1 is present in the mesenchyme[52,53]. ETV1 alone controls heart morphogenesis and maturation of murine and hPSC-derived cardiomyocytes[54,55]. Moreover, Etv1 downregulation

retracts the dendritic process and arrests granule cells in the immature status in mice[56]. All three ETVs are expressed in ESCs and during reprogramming to iPSCs. Etv1 expression appears to be negligible for mouse ESCs, whereas Etv4 and Etv5 maintain the naïve pluripotent state and the transition to primed pluripotency in mESCs[57,58]. However, ETVs' role in hESCs and pancreatic differentiation has not been reported.

Here, we characterize the role of the ETV genes in hESCs and cell fate determination during gastrulation (for ETV1, ETV4, and ETV5 genes) and pancreatic in vitro differentiation, with the focus on ETV1 gene, as ETV4 and ETV5 expression is not detected in human pancreatic epithelium. Specifically, ETV loss alters the cytoskeleton, enhances cell–cell and cell-ECM adhesion via PI3K/AKT signaling, and hinders proper differentiation into three germ layers. Moreover, ETV1 knock-out (KO) impairs cell–cell and cell-ECM adhesion during in vitro pancreatic differentiation, markedly reducing EP formation. Collectively, our results suggest that ETV1, ETV4, and ETV5 are critical regulators of mechanosignaling in hPSCs and pancreatic cells, influencing multilineage differentiation. Ultimately, controlling adhesion during hPSC differentiation will allow the precise guidance of hPSCs toward desired lineages and recapitulation of the specific 3D tissue architecture.

## Results

### Deletion of ETV1, ETV4, and ETV5 genes in human pluripotent stem cells

The PEA3 subfamily members, ETV4 and 5 have been identified as regulators of naïve pluripotency in mouse ESCs[57,58]. Conversely, ETV1, a closely related family member, is expressed in the mouse ESCs and e3.5–e4.5 epiblast cells, where it appears to play a minor role[57,58]. To investigate ETV expression in hPSCs, we initially analyzed publicly available single-cell RNA sequencing datasets, which revealed the expression of ETV1, ETV4, and ETV5 mRNA in OCT3/4+ cells (Supplementary Fig. 1A). Using immunofluorescence staining and flow cytometry, we corroborated the presence of ETV1, ETV4, and ETV5 proteins in OCT3/4+, NANOG+, or KLF4+ hPSCs (Fig. 1A–C).

To investigate the role of ETV genes in hPSCs and during in vitro differentiation, we employed a CRISPR/Cas9 approach to knock out the ETV1 gene in hPSCs. Specifically, we targeted ETV1 exon 4, encoding the PEA3 domain necessary for DNA binding and induced a deletion that results in a premature stop codon (Fig. 1D). To assess the gene editing efficiency in the hPSC population, we deconvoluted DNA sequencing data of targeted loci using Inference of CRISPR Edits (ICE) tool. The ICE analysis of ETV1 exon 4 or 9 targeted samples revealed -80% alleles with frameshift-inducing indels in either exon (Supplementary Fig. 1B). To generate clonal KO hPSC lines, we picked individual colonies, and using genomic PCR with primers flanking the targeted regions (Supplementary Fig. 1C), and Sanger DNA sequencing, we identified clones with desired null mutations in the ETV1 gene. Western blot analysis confirmed the absence of ETV1 protein in ETV1 KO hPSCs (Fig. 1E). We further refer to the ETV1 KO hPSC line with a stop codon in exon 4, as "KO". To generate an independent ETV1 loss-of-function model in hPSCs, we targeted exon 9 of ETV1, which also encodes the PEA3 domain, using a different sgRNAs, resulting in the deletion and premature stop codon in exon 9 (Supplementary Fig. 1D) as was confirmed by genomic PCR, with primers flanking the targeted region of exon 9 (Supplementary Fig. 1E), and DNA sequencing. We will henceforth refer to the ETV1 KO exon 9 hPSC line as "KO2".

In hPSCs, several ETVs are coexpressed (Fig. 1B and Supplementary Fig. 1A), suggesting potential redundancy or a cooperative role in controlling pluripotency and differentiation. In ETV1 KO hPSCs, we observed the upregulation of ETV4 and ETV5 mRNAs, suggesting a compensatory response after ETV1 deletion (Fig. 1F). Consequently, we pursued the generation of triple ETV1/ETV4/ETV5 KO (tKO) by targeting ETV4 and ETV5 genes in the ETV1 KO hPSC line. We induced frameshift deletions in exons 2–3 of ETV4 (Fig. 1G), and exons 3–4 of ETV5

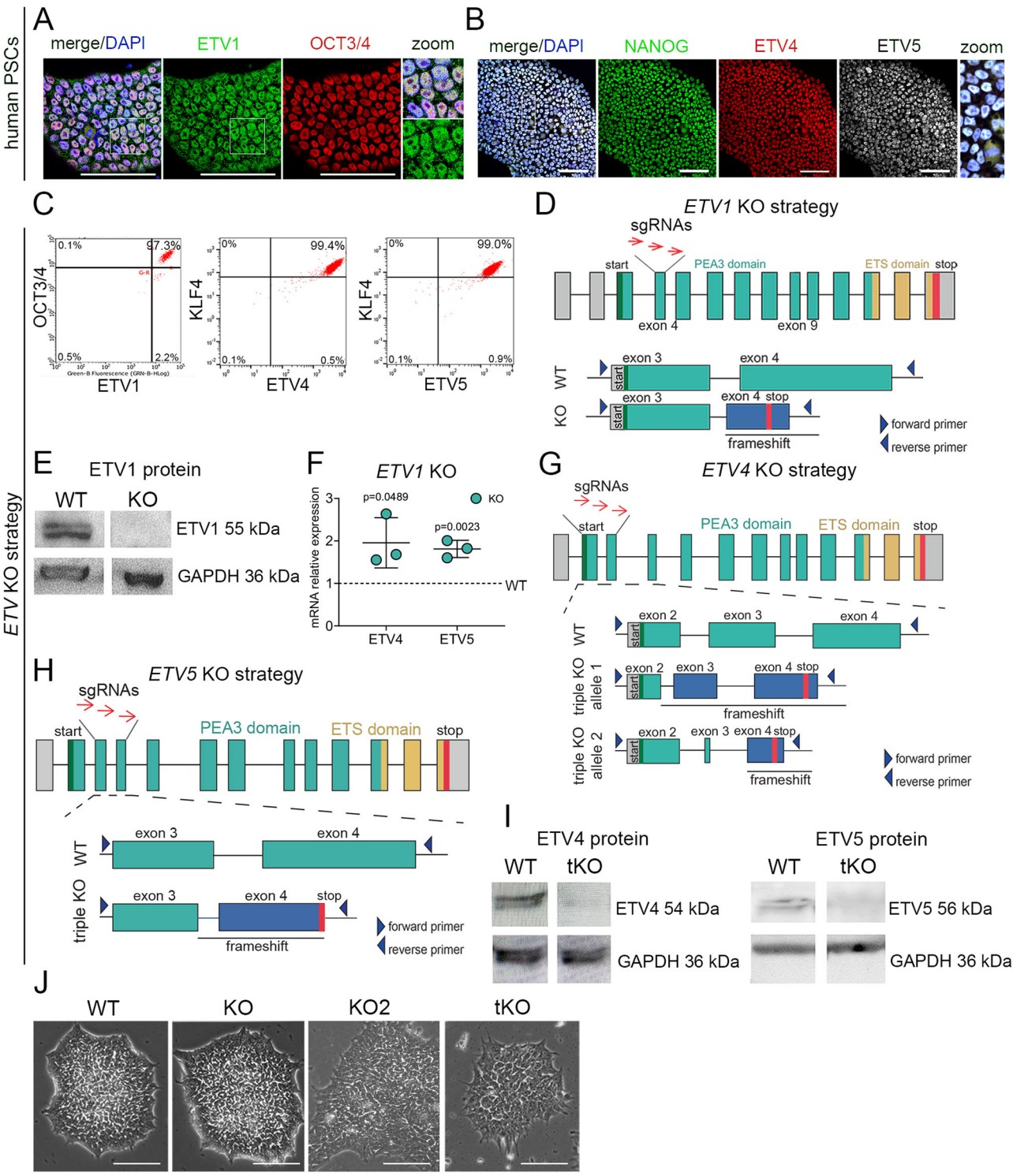

(Fig. 1H) and identified clones with desired mutations using genomic PCR (Supplementary Fig. 1F) and Sanger DNA sequencing. The deletion of ETV4 and ETV5 proteins in tKO hPSCs was validated by western blot analysis (Fig. 1I). In the initial assessment, the colony morphology of *ETV1* KO and KO2 hPSCs was comparable to the parental hPSC line, while tKO colonies exhibited a different, looser, and jagged structure (Fig. 1J). Notably, the expression of pluripotency markers, SOX2, OCT3/4, and NANOG, was comparable in KO, tKO, and parental WT hPSCs (Supplementary Fig. 1G, H). In summary, to investigate the ETV role in hPSCs, we generated several clonal hPSC lines: two independent ETV1 loss-of-function lines, KO and KO2; and ETV1, ETV4 and ETV5 triple deletion line, tKO.

## Deletion of *ETV* genes increases hPSC confluency via enhanced cell–matrix and cell–cell adhesion

Using live-cell imaging over a three-day-long culture, we observed differences in total cell growth between ETV-depleted and WT hPSCs. Specifically, the cell confluency was significantly higher in KO and KO2 (by 52% and 42%, respectively), and tKO (by 74%) compared to WT hPSCs, irrespective of the tested seeding densities. Intriguingly, these disparities in confluency were evident as early as 2 h post-plating (Supplementary Fig. 2A–C). The heightened confluency might result from enhanced proliferation, initial cell attachment, and/or increased cell size. We, therefore, investigated each of these possibilities. The expression of the widely used proliferation marker, phospho-histone 3

**Fig. 1 | Deletion of *ETV1*, *ETV4*, and *ETV5* genes in hPSCs. A** Representative images of immunofluorescence staining show the ETV1 protein (green) presence in OCT3/4 (red) positive cells. DAPI was used to label nuclei (blue). Protein coexpression is presented on the right in enlarged inserts. *N* = 3 biological replicates. Scale bar = 100 µm. **B** Representative immunofluorescence staining shows the presence of ETV4 (red), and ETV5 (gray), in NANOG+ (green) cells. DAPI was used to label nuclei (blue). Protein coexpression is presented on the right in the enlarged insert. *N* = 3 biological replicates. Scale bar = 100 µm. **C** Representative flow cytometry analysis shows the coexpressions of: ETV1 and OCT3/4 proteins, and ETV4, ETV5, and KLF4 proteins in hPSCs; 97–99% of hPSCs expressed OCT3/4 and ETVs, while 99.4% and 99% of KLF4 + hPSCs expressed ETV4 and ETV5, respectively. *N* = 3 biological replicates. **D** Strategy to knock out *ETV1* gene in hPSCs using CRISPR/ Cas9 approach. Three different sgRNAs (red arrows) targeting exon 4 of the *ETV1* gene were co-transfected into hPSCs, and pretreated with doxycycline to induce Cas9 expression, resulting in a 127-nucleotide deletion and a premature stop codon. Untranslated regions (gray), PEA3 domain (turquoise green), ETS domain (dark yellow), STOP codon (red). Primer positions are shown by arrowheads. **E** ETV1 protein absence in KO compared to WT hPSCs, demonstrated by western blotting. An antibody against GAPDH was used as a loading control. **F** The relative expression level of *ETV4* and *ETV5* was higher in ETV1 KO compared to WT cells (baseline set at 1), shown by qPCR. The data are presented as the means ± SDs. A two-sided student's *t*-test was used to determine the *p*-values shown on the graph. *N* = 3 biological replicates. **G, H** Strategy to generate *ETV4, ETV5,* and *ETV1* triple KO (tKO) in hPSCs. Three sgRNAs (red arrows) targeting exons 2 and 3 of *ETV4* (**G**), and exons 3 and 4 of *ETV5* (**H**) were co-transfected into *ETV1* KO hPSCs, leading to deletions and premature stop codons. Untranslated regions (gray), PEA3 domain (turquoise green), ETS domain (yellow), STOP codon (red). Primer positions are shown by arrowheads. **I** Western blot analysis of ETV4 or ETV5 proteins in tKO and WT hPSCs. An antibody against GAPDH was used as a loading control. *N* = 2 biological replicates. **J** Representative bright-field images of WT, KO, KO2, and tKO hPSC colonies. Scale bar = 200 µm. *N* = 5 biological replicates.

(pHH3), was assessed, revealing that at 72 h post-seeding, the number of pHH3+ KO and tKO cells was comparable to that of WT hPSCs (Supplementary Fig. 2D). We confirmed this observation, by assessing Ki67+ WT and KO hPSCs (Supplementary Fig. 2E). A closer examination of the initial hours of KO, KO2, and tKO hPSC outgrowth revealed that, despite equal initial seeding cell density all knockout lines showed increased cell confluency (Fig. 2A, B), cell number (Fig. 2A, C) and cell spreading (measured as ratio of confluency to the total cell number) (Fig. 2A, D) as early as 2 h after cell seeding (Fig. 2E–G). These results suggest that the enhanced initial attachment and spreading observed in *ETV* KO, and tKO lines may be responsible for the increased confluency. Finally, flow cytometry analysis demonstrated increased forward scatter and side scatter, indicating that KO and tKO cells are larger (Supplementary Fig. 2F).

We next used a quantitative assay based on crystal violet binding to DNA and proteins in adherent cells to investigate the hPSC attachment to different coatings. We plated $1.5 \times 10^4$/cm$^2$ WT, KO, and tKO cells either in standard hPSC media (ROCKi-) or in media supplemented with Rho-kinase inhibitor (ROCKi+) on various ECM proteins (Fig. 2H). ROCKi prevents anoikis, commonly occurring during passaging when hPSC colonies are disrupted to single cells[59]. At 24 h post-seeding, KO and tKO cells plated on hPSC-specific ECM substrates, such as Geltrex or Vitronectin, exhibited significantly higher initial attachment regardless of ROCKi presence (Fig. 2I). We also observed a significant increase in KO and tKO cell confluency when hPSCs were seeded on other ECM substrates, including Laminin V or Fibronectin, except for tKO seeded on Laminin V in ROCKi-deficient media (Fig. 2I). These findings suggest that ETV1, ETV4 and ETV5 may control cell adhesion regardless of the cell culture conditions or ECM substrate. Finally, to ascertain whether the cell culture format influenced our observations, we measured the 3D clustered density of KO or tKO cells using physical cytometry. Remarkably, we observed a higher density of clusters composed of KO or tKO hPSCs compared to WT (Fig. 2J).

Subsequently, we investigated whether ectopic *ETV1* expression (OE) in WT hPSCs could induce the opposite effect - a decrease in hPSC adhesion. To this end, we established hPSC line with a doxycycline-inducible *ETV1* overexpression (OE) based on the *piggy-Bac* transposon[60] and observed a strong upregulation of *ETV1* expression and a loosened colony morphology upon doxycycline treatment (Fig. 2K). The *ETV1* OE hPSC showed decreased colony integrity, with cells loosely arranged on the edges, pointing to a decline in hPSC adhesion (Fig. 2K). Further, we overexpressed ETV1 in the KO hPSCs to obtain a "rescue model". Next, we measured the levels of known adhesion proteins - CDH1 and integrin α 5 in ETV1 OE cell lines (WT_OE and KO_OE) and observed decreased levels of both adhesion proteins compared to WT hPSCs (Fig. 2L, and Supplementary Fig. 2G). Additionally, we set up a system for pulsing hPSC colonies with *ETV1* overexpression at low frequency to trace single

cells, using a transient, plasmid-based approach. Employing live-cell imaging, we tracked the transfected cells every 2 h, commencing from the time of lipofection. In cells transfected with empty backbone plasmid, we saw GFP positivity at 24 h post-transfection. In contrast, cells receiving *ETV1*-cDNA-GFP plasmid were GFP+ at t = 0 h, but within a short timeframe (~4 h) the GFP+ signal started to diminish, either due to cell death or a detachment from the plate (Supplementary Fig. 2H). Quantitative analysis of GFP+ cells revealed that hPSCs transfected with empty plasmid remained attached for a longer duration compared to those with *ETV1* ectopic expression. For instance, ~50% of hPSCs overexpressing *ETV1* detached after 8 h, while almost 100% of cells transfected with the backbone plasmid remained alive and attached (Supplementary Fig. 2I). We, therefore, concluded that *ETV1* overexpression impaired hPSC attachment, leading to extensive cell loss. Collectively, these data suggest that ETV1, either alone or in combination with ETV4 and ETV5, controls cell adhesion and attachment to different ECM substrates.

### *ETV1* deletion causes the upregulation of cell–cell and ECM adhesion-associated genes and the downregulation of pluripotency- and cell signaling-associated genes in hPSCs

We investigated the molecular changes associated with ETV1 deletion in hPSCs, both independently or in conjunction with ETV4 and ETV5 null mutations, using RNA sequencing (RNA-seq). The analysis revealed several hundreds of differentially expressed genes (DEGs) across three comparisons: ETV1 KO vs. WT, tKO vs. WT, and tKO vs. ETV1 KO (Fig. 3A–C, Supplementary Fig. 3A, and Supplementary Data 1). In the ETV1 KO vs. WT comparison, the top DEGs included Integrin Subunit Alpha 5 (*ITGA5*), N-cadherin (*CDH2*), Caveolin-1 (*CAV1*), and Collagen Type IV Alpha 6 Chain (*COL4A6*), all of which showed increased expression, while Frizzled-8 (*FZD8*) or KLF transcription factor 4 (*KLF4*) were downregulated. In the tKO vs. WT dataset, genes such as Vimentin (*VIM*), Serpin Family E Member 1 (*SERPINE1*), and Laminin Subunit Alpha 2 (*LAMA2*) displayed increased expression, mirroring trends observed in the tKO vs. ETV1 KO comparison.

Hierarchical clustering of DEGs, along with analysis of enriched biological processes and KEGG/WikiPathways signaling pathways, revealed terms related to cell–cell adhesion, focal adhesion, cytoskeletal regulation, and PI3K/AKT signaling pathways in both ETV1 KO and tKO samples (Fig. 3D, E, G, H, and Supplementary Fig. 3B, C), supporting our observation of altered adhesion in ETV-deficient hPSCs. Notably, the tKO vs. single KO comparison showed further upregulation of genes associated with focal adhesion, PI3K/AKT/ mTOR pathway, cell–cell adhesion, and collagen organization (Fig. 3F). Genes related to adhesion, including *CDH2, CDH3, CDH24,* and *COL6A1*, exhibited more pronounced dysregulation in the tKO compared to ETV1 KO, with both upregulation and downregulation

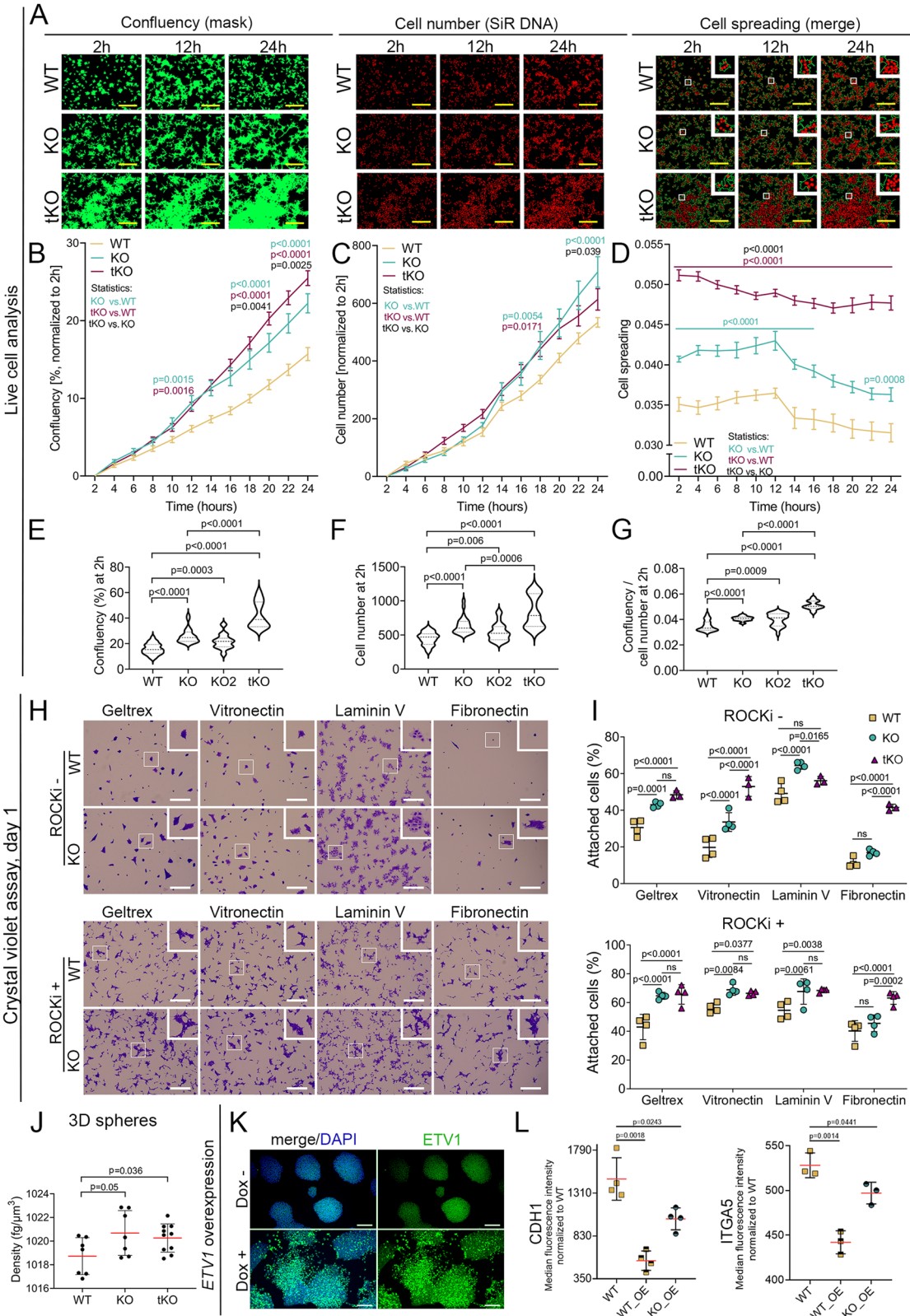

observed. This suggests that the deletion of ETV4 and ETV5 amplifies the effect of ETV1 KO on the adhesion-related processes (Fig. 3G, H). Interestingly, we identified a small cohort of adhesion-related genes, whose expression depended solely on ETV1, but not on other ETVs (Supplementary Fig. 3C).

The enrichment terms for downregulated genes in KO and tKO were linked to cell–cell contact and cytoskeleton, but also WNT, BMP4, Notch, and TGFβ pathways, and signaling to regulate pluripotency

(Fig. 3D, E, G, H, and Supplementary Fig. 3B, C), suggesting the changes in the ETV KO hPSC differentiation potential. Interestingly, a more profound effect on cell fate commitment was observed in the single KO compared to the tKO, as revealed by the hierarchical clustering of DEGs related to the ectoderm, endoderm, and mesoderm differentiation (Fig. 3G, H, and Supplementary Fig. 3C). To further investigate the impact of ETVs on adhesion and pluripotency, we performed ATAC-seq of KO, tKO and WT hPSCs. Analysis of the differential chromatin

**Fig. 2 | Increased adhesion of *ETV1* and *ETV1*/*ETV4*/*ETV5* deficient hPSCs.**
**A** Growth dynamics of KO, tKO, and WT hPSCs over a 24 h culture monitored by live-cell imaging. Left - representative images of cell confluency at 2, 12, and 24 h culture, marked by green mask. Middle-representative images of cell number at 2, 12, and 24 h culture, marked by nuclei marker SiR DNA (red). Right - representative images of cell spreading (quantified as the ratio of confluency to cell number) at 2, 12, and 24 h culture. A green line marks confluency and nuclei are marked by SiR DNA in red. Scale bar = 400 μm. **B** Quantification of WT (yellow), KO (green), and tKO (maroon) hPSC confluency over a 24 h culture. tKO cells show the highest confluency, a 166% increase compared to WT cells. $N = 3$ biological replicates. At 24 h, for KO vs. WT and tKO vs. WT, $p < 0.0001$, for tKO vs. KO, $p = 0.0025$.
**C** Quantification of cell number over a 24 h culture. KO (green) and tKO (maroon) show an increase in the cell number compared to WT cells (yellow). The increase in cell number was the most pronounced for tKO cells (66% increase in comparison to WT) and by 36% for KO hPSCs. $N = 3$ biological replicates. At 24 h, for KO vs. WT, $p < 0.0001$, for tKO vs. KO, $p = 0.039$; at 18 h, for tKO vs. WT, $p = 0.0171$.
**D** Quantification of cell spreading (confluency/cell number) over a 24 h culture. KO (green) and tKO (maroon) show an increase in cell spreading compared to WT cells (yellow). $N = 3$ biological replicates. At 24 h, for KO vs. WT, $p = 0.0008$, for tKO vs. WT and tKO vs. KO, $p < 0.0001$. **E** Confluency quantification at 2 h post-seeding (corresponds to **A**). The highest increase in cell confluency was noted for tKO followed by KO and KO2 in comparison to WT hPSCs. $N = 3$ biological replicates. For KO vs. WT, tKO vs. WT, tKO vs KO, $p < 0.0001$, for KO2 vs. WT, $p = 0.0003$. **F** Cell

number quantification at 2 h post-seeding (corresponds to **A**). The highest increase in cell number was noted for tKO followed by KO and KO2 hPSCs. $N = 3$ biological replicates. For KO vs. WT and tKO vs. WT, $p < 0.0001$, for KO2 vs. WT, $p = 0.006$, tKO vs. KO, $p = 0.0006$. **G** Quantification of cell spreading (confluency/cell number) at 2 h post-seeding (corresponds to **A**). The most pronounced increase in cell spreading was noted for tKO hPSCs followed by KO and KO2. $N = 3$ biological replicates. For KO vs. WT, tKO vs. WT, tKO vs KO, $p < 0.0001$, for KO2 vs. WT, $p = 0.0009$. **H** Representative crystal violet staining images of cells cultured in the absence (top panel) or presence (bottom panel) of ROCK inhibitor (ROCKi) 24 h after hPSC seeding. The same number of WT and KO cells were seeded on different surface coatings, as indicated. Scale bar = 400 μm. **I** Quantification of crystal violet staining shows an increase in the attachment of KO (green) and tKO (maroon) compared to WT (yellow) cells, on all tested surface coatings in the absence (top panel) and presence (bottom panel) of ROCKi. $N = 4$ biological replicates. **J** Physical cytometer analysis of KO, tKO, and WT hPSC spheres demonstrated enhanced density of KO and tKO compared to WT. $N = 3$ biological replicates. **K** Representative images of immunofluorescence staining show ETV1 (green) overexpression in hPSCs induced by 24 h doxycycline treatment (Dox+), compared to untreated (Dox-) cells. **L** Quantification of adhesion protein levels CDH1 (left) and ITGA5 (right) in WT_OE and KO_OE hPSCs after induction of *ETV1* overexpression. CDH1, $N = 4$ biological replicates; ITGA5, $N = 3$ biological replicates. For plots **B–G, I, J**, and **L**, a one-way ANOVA for multiple comparisons was used to determine the *p*-values shown on the graph. The data are presented as means ± SDs.

accessibility regions (DARs) revealed 28 and 242 distinct DARs in KO and tKO, respectively, compared to WT cells (adjusted p-value < 0.1). We observed enrichment of the ATAC-seq peaks in the regions encoding genes responsible for adhesion (i.e., *ITGAX*) and pluripotency (i.e., *VAV1*) in the ETVs KO compared to WT hPSCs, which further corroborated our findings (Fig. 3I, J). Together, the loss of ETVs in hPSCs leads to dysregulation of adhesion and downregulation of multiple signaling pathways regulating pluripotency and cell fate.

### ETV1 transcriptional targets in hPSCs regulate cell–cell and cell–ECM adhesion

To identify *ETV1* direct targets in hPSCs, we performed an in silico analysis followed by ChIP-qPCR validation. Using the general ETV motif matrix, chosen due to the indistinguishable binding sequences of *ETV1*, *ETV4*, and *ETV5*, we conducted enrichment analysis with Pscan software. DEGs shared between *ETV1* KO and tKO hPSCs served as input, and we applied the following parameters: a promoter region spanning −950 to +50 bp from the transcription start site (TSS), the *ETV* matrix motif from the Jaspar database, and a z-score above 0.8 (where 1 is the highest score). This led to the identification of 260 upregulated and 657 downregulated genes with a potential *ETV* binding motif (Supplementary Fig. 3D). Next, we subjected these putative ETV-regulated DEGs to KEGG, Panther and BP analysis, revealing the enrichment of terms related to focal adhesion and the PI3K/AKT signaling pathway for upregulated genes, and pluripotency regulation for downregulated genes (Supplementary Fig. 3E, F). Examples of genes with a potential *ETV* binding motif include those associated with cell adhesion, such as *ITGA5*, Vinculin (*VCL*), *PDGFRB*, and *COL4A6*, and those involved in pluripotency regulation, such as *JUN*, *CDKN1A* (Supplementary Fig. 3G). Interestingly, ETV1 immunoprecipitation followed by western blot analysis, showed ETV1 presence in both the chromatin-bound (nuclear) fraction and soluble protein (cytoplasmic) fraction in hPSCs, with a substantial proportion of ETV1 residing in the unbound state (Fig. 3K). Finally, we performed ChIP-qPCR, which demonstrated enhanced *ETV1* binding to the promoter regions of selected genes (Fig. 3L), confirming the accuracy of in silico predictions.

### Elevated levels of adhesion-associated proteins in *ETV1* KO and *ETV1*, *ETV4*, and *ETV5* tKO hPSCs

To further investigate the changes in adhesion observed in *ETV1* KO hPSCs, we examined the protein levels of four adhesion-related genes: *ITGA5*, *VCL*, Paxillin (*PXN*), and E-cadherin (*CDH1*), all of which showed

increased transcript levels in *ETV1* KO. ITGA5 acts as a fibronectin receptor in conjunction with integrin 1β, facilitating cell-ECM attachment[61,62]. *ETV1* deletion in hPSCs led to a ~25% increase in ITGA5 protein expression (Fig. 4A and Supplementary Fig. 4A). Within cells, integrin receptors bind to talin or PXN, which, in turn, link to actin via VCL, thereby connecting the ECM with the intracellular cytoskeleton. Notably, we observed elevated expression of VCL and PXN proteins (both by ~30%) in *ETV1* KO (Fig. 4B–D and Supplementary Fig. 4B–E). VCL can also facilitate cell–cell junctions by binding to α- and β-catenin, which are crucial for stabilizing CDH1 at the cell surface. CDH1, a transmembrane glycoprotein mediating cadherin-dependent cell–cell adhesion, showed a 50% increase in levels in *ETV1* KO hPSCs, as revealed by immunofluorescence staining and quantification (Fig. 4D, E, and Supplementary Fig. 4D). The abundance of CDH1 and PXN proteins was also increased in KO2 compared to WT hPSCs, by 50% and 20%, respectively (Supplementary Fig. 4F). Furthermore, VCL, CDH1, and PXN protein levels were elevated in tKO (by 30%, 40% and 130%, respectively; Fig. 4C, D and Supplementary Fig. 4C, D). CDH1 and PXN protein distribution was also affected in KO and tKO hPSC colonies (Fig. 4E and Supplementary Fig. 4G). The morphology of the tKO colony was disrupted, with a higher presence of gaps (Fig. 4F and Supplementary Fig. 4H). Finally, we observed dysregulation of actin cytoskeleton organization in tKO compared to WT, marked by altered arrangement of F-actin filaments, likely resulting from changes in cell tension and mechanosignaling. This difference in F-actin structure was more pronounced at colony edges (Fig. 4G) than in the colony center (Supplementary Fig. 4I) in tKO compared to WT. Collectively, these results suggest that biophysical features, including hPSC adhesion to the ECM and other cells, are regulated by ETV1, ETV4, and ETV5.

### PI3K/AKT signaling controls hPSC adhesion

We next asked whether modulating ETV1-dependent signaling pathways could regulate hPSC adhesion. RNA-seq analysis of DEGs in *ETV1* KO hPSCs showed upregulation of the PI3K/AKT signaling pathway (Fig. 3D), which is known to regulate cell attachment and spreading[63]. While the total AKT protein levels remained unchanged between KOs and WT hPSCs, phosphorylated AKT (at Ser473) exhibited up to a 3.4-fold increase in tKO, indicating heightened AKT activation (Fig. 4H and Supplementary Fig. 4J). To further explore this, we modulated the PI3K/AKT pathway, using two inhibitors, PI-103 and Torin2, and an activator - insulin (Fig. 4I–L). *ETV1* KO and WT hPSCs were seeded in the presence of varying concentrations of these compounds, with

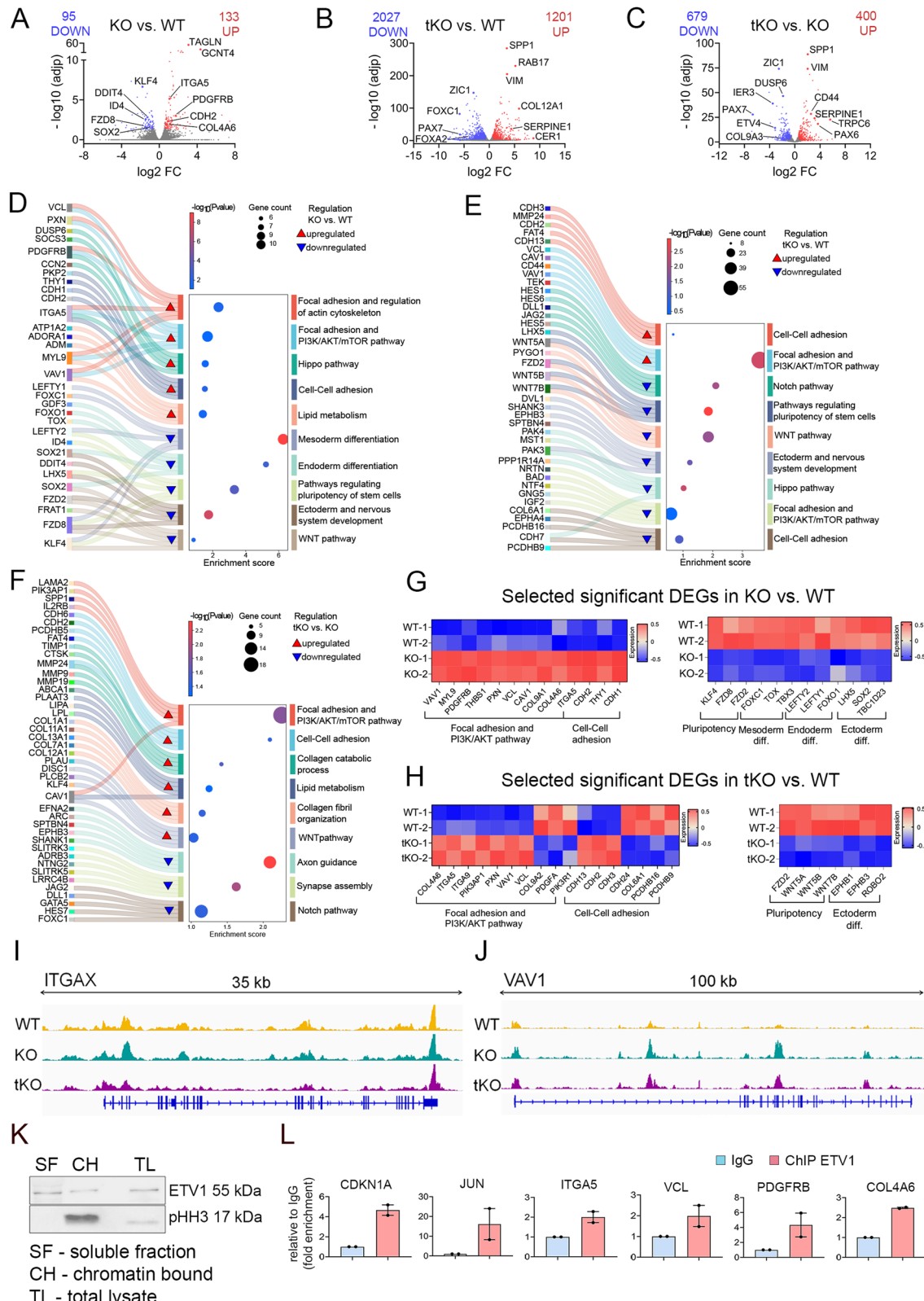

DMSO serving as the vehicle control, and cell attachment was monitored during the initial 24 h using live-cell imaging. At 24 h, we observed a decrease in the confluency of *ETV1* KO and WT hPSCs treated with either PI3K/AKT inhibitor compared to the controls, as measured by crystal violet staining (Fig. 4J, K, and Supplementary Fig. 4K, L). Notably, *ETV1* KO treated with 15 nM Torin2 showed similar confluency levels to untreated WT hPSCs (Fig. 4K), suggesting that

PI3K/AKT signaling inhibition rescued the enhanced adhesion caused by *ETV1* deletion. Conversely, insulin led to increased confluency in WT hPSCs but not in *ETV1* KO, suggesting that *ETV1* ablation saturated PI3K/AKT pathway activation (Fig. 4L). We then investigated whether confluency changes in *ETV1* KO hPSCs caused by PI3K/AKT pathway modulation were associated with differential expression of adhesion-related proteins. Prolonged (72 h) Torin2 treatment led to decreased

**Fig. 3 | RNA sequencing and CHIP-qPCR analysis reveal genes and pathways regulated by ETV1, ETV4, and ETV5 transcription factors in hPSCs. A** A volcano plot was generated to visualize differentially expressed genes between KO and WT hPSCs. Red dots indicate 133 upregulated genes, and blue dots indicate 95 downregulated genes, showing log2 fold change (log2 FC) and −log10 adjusted *p*-value (−log10 adjp). **B** A volcano plot was generated to visualize differentially expressed genes between tKO and WT hPSCs. Red dots indicate 1201 upregulated genes, and blue dots indicate 2027 downregulated genes, showing log2 fold change (log2 FC) and −log10 adjusted *p*-value (−log10 adjp). **C** Volcano plot of differentially expressed genes between tKO and KO hPSCs shows log2 fold change (log2 FC) and −log10 adjusted *p*-value (−log10 adjp) for 679 downregulated (blue) and 400 upregulated (red) genes. **D** The top-ranked enriched terms (adjusted p-value ≤ 0.05) from the KEGG, WikiPathways, and Biological Processes databases were identified among the differentially expressed genes between KO and WT hPSCs. Enriched functional terms associated with upregulated and downregulated genes in KO compared to WT hPSCs are marked by red and blue triangles, respectively. **E** The top-ranked enriched terms (adjusted p-value ≤ 0.05) from the KEGG, WikiPathways, and Biological Processes databases among the differentially expressed genes between tKO and WT hPSCs. Enriched functional terms associated with upregulated and downregulated genes in tKO compared to WT hPSCs are marked by red and blue triangles, respectively. **F** The top-ranked enriched terms (adjusted p-value ≤ 0.05) from the KEGG, WikiPathways, and Biological Processes databases among differentially expressed genes between tKO and KO hPSCs. Enriched functional terms associated with upregulated and downregulated genes in tKO compared to WT hPSCs are marked by red and blue triangles, respectively. **G** Heatmaps demonstrate the top significant differentially expressed genes in KO compared to WT associated with cell−cell and cell-ECM adhesion, PI3K/AKT pathway (left panel), or pluripotency and cell fate (right panel). Diff. – differentiation. **H** Heatmaps demonstrate the top significant differentially expressed genes in tKO compared to WT associated with cell−cell and cell-ECM adhesion, and PI3K/AKT pathway (left panel) or pluripotency and cell fate (right panel). Diff. – differentiation. **I** ATAC-seq tracks highlight the *ITGAX* locus in WT (yellow), KO (green), and tKO (maroon) hPSCs. The peaks represent normalized and combined biological replicates ($N = 2$). **J** ATAC-seq tracks highlight the *VAV1* locus in WT (yellow), KO (green), and tKO (maroon) hPSCs. The peaks represent normalized and combined biological replicates ($N = 2$). **K** Western blot demonstrates the ETV1 presence in both the chromatin-bound (CH) and soluble protein fraction (SF) in WT hPSCs. pHH3 was used as a positive control of fractionation, and total cell lysate (TL) was used as the positive control for ETV1. $N = 2$ biological replicates. **L** ChIP-qPCR analysis (antibody against ETV1) confirmed ETV1 binding to the promoter region of selected differentially expressed genes connected to cell adhesion, i.e., *ITGA5, VCL, PDGFRB,* and *COL4A* or pluripotency, i.e., *JUN* and *CDKN1A*. IgG – control (blue); ChIP ETV1 – target (red). Data are presented as the mean ± SDs. $N = 2$ biological replicates. For plots (**A**–**C**), the DESeq2 package and the Wald test were used to determine significance. DAVID online software was used for biological term enrichment analysis of the DEGs, with the Fisher Exact statistics to list annotation terms and their associated genes. The *p*-values are adjusted for multiple comparisons using the Benjamini and Hochberg approach. Genes with non-significant change in expression are depicted in gray (−log10 (adjp) <2; log2 FC between 0.5 and −0.5). Highlighted dots indicate selected genes in the enriched pathways, pictured in plots (**D**–**F**).

---

levels of CDH1 and PXN proteins in *ETV1* KO and WT hPSCs compared to their respective untreated or DMSO-treated controls (Fig. 4M−O and Supplementary Fig. 4M). In contrast, insulin treatment increased CDH1 and PXN protein levels exclusively in WT cells, but not in *ETV1* KO cells (Fig. 4M−O). Collectively, these data indicate that *ETV* genes regulate hPSC adhesion through the PI3K/AKT signaling pathway.

### *ETV1* KO hPSCs show a higher propensity to differentiate into mesoderm and endoderm

As we identified a cohort of DEGs related to differentiation or pluripotency in KO or tKO (Fig. 3D−H), we next sought to understand how the loss of *ETV* genes impacts hPSC differentiation potential. To address this, we first employed a micropatterning approach to generate organized embryo-like structures[64] referred to as gastruloids, which are designed to mimic the development and architecture of the three germ layers observed in vivo (Fig. 5A). We treated hPSCs confined to micropatterned colonies with bone morphogenetic protein 4 (BMP4) to induce the organized differentiation, with extraembryonic-like cells emerging on the outer edge of the colony, ectodermal cells in the colony center, and mesendodermal cells forming the layers in between. Despite being constructed in 2D, this multicellular pattern exhibited a radial spatial organization of different germ layers consistent with a gastrulating embryo (Fig. 5B). Control ESI017 hPSCs showed identical patterns to those we have previously published with this cell line[65], while control HUES8 and HUES8-iCas9 cells showed similar patterns but with enhanced BRA expression and decreased SOX2 expression, possibly indicating a slight propensity for mesendoderm differentiation in this line[66]. In *ETV1* KO, BMP4 treatment led to completely disrupted radial organization of all germ layers, as demonstrated by staining with BRA, a mesodermal/endodermal marker and SOX2 (an ectodermal marker) (Fig. 5B, C and Supplementary Fig. 5A). In tKO, we observed a drastically altered pattern, characterized by increased layers of extraembryonic-like cells marked by ISL LIM homeobox 1 (ISL1), moving toward the center of the gastruloid, and near the disappearance of the ectodermal layer marked by SOX2 (Fig. 5B, C and Supplementary Fig. 5A).

Since mesendoderm formation was disrupted in KO and tKO gastruloids, we next investigated definitive endoderm (DE) formation in ETV-deficient hPSCs. Using a standard protocol with

100 ng/ml of Activin A and 3 μM of CHIR99021 (a WNT signaling agonist), we observed no significant differences in the percentage of SOX17+ cells between KO and WT hPSCs, indicating similar differentiation efficiency. However, tKO cells showed a small decrease in the percentage of SOX17+ (Fig. 5D). We hypothesized that these strong pro-endoderm conditions might mask potential defects in ETV KO hPSCs. Therefore, we conducted directed endoderm differentiation using two approaches: (i) a suboptimal protocol with 1/3 the normal dose of Activin A, with or without CHIR99021 (Fig. 5E), and (ii) spontaneous differentiation into three germ layers via embryoid body (EB) formation in media lacking specific signals to maintain pluripotency or direct cell fate (Fig. 5F). Reducing the Activin A concentration decreased DE efficiency in all cell lines (Supplementary Fig. 5B). However, no significant differences were observed between WT, KO, and tKO under these suboptimal conditions. In contrast, removing the WNT activator significantly decreased DE formation in KO (60%) and tKO (-10%) cells compared to WT (-20%) cells (Fig. 5E), suggesting that ETV genes regulate DE formation through the WNT pathway.

During spontaneous differentiation in the absence of any lineage-specific growth factors, we observed changes in the *ETV1* KO EB morphology by day 8 or 10, suggesting changes in cell−cell or cell-ECM adhesion (Supplementary Fig. 5C). qPCR analysis revealed higher expression of a cohort of genes associated with endoderm and mesoderm in ETV1 KO cells, including GATA Binding Protein 4 (*GATA4*), Motor Neuron and Pancreas Homeobox 1 (*MNX1*), SRY-Box Transcription Factor 17 (*SOX17*), Homeobox A1 (*HOXA1*), Meis Homeobox 2 (*MEIS2*), and Platelet Derived Growth Factor Receptor Alpha (*PDGFRA*). This suggests a higher propensity for mesoendoderm differentiation (Fig. 5G). Conversely, *ETV1* KO cells showed less efficient ectodermal differentiation (Fig. 5G). Together, *ETV* genes are essential regulators of hPSC cell fate decisions, especially during early differentiation.

### *ETV1* KO hPSCs fail to robustly differentiate into pancreatic endocrine progenitors

In our final analysis, we used directed differentiation to generate pancreatic cells from *ETV1* KO hPSCs to gain deeper insights into their differentiation potential. This focus on pancreatic lineage was

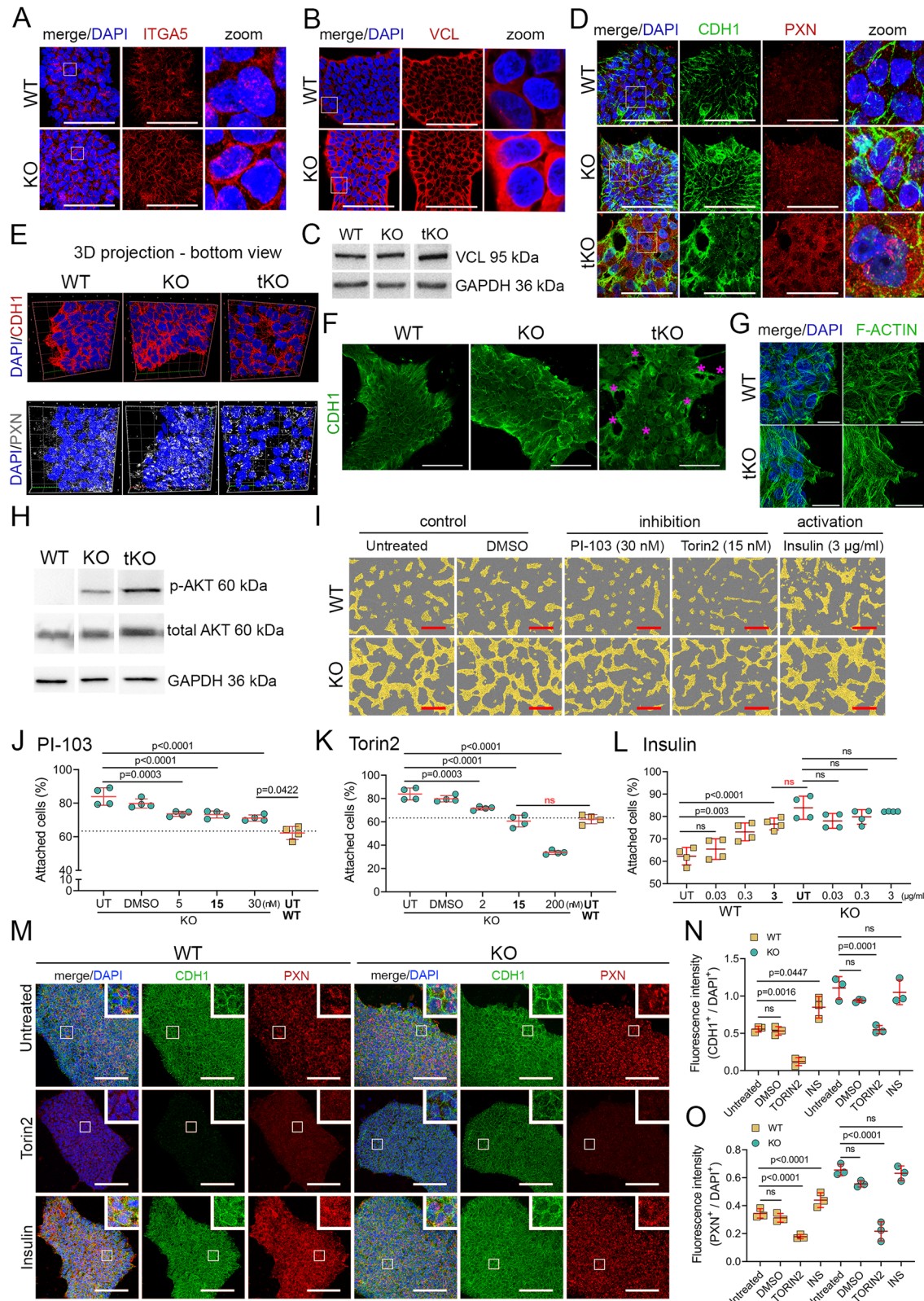

informed by the *ETV1* expression in the EP subpopulation during the remodeling and budding of epithelial pancreatic cords in the developing murine pancreas[42]. Moreover, we confirmed ETV1 protein expression in PDX1+ pancreatic epithelium and in Chromogranin A (CHGA)+ endocrine cells in the human fetal pancreas at 10.6- and 13-week post-conception (wpc) (Fig. 5H). These observations prompted us to investigate whether ETV1 regulates adhesion during human

pancreatic development. WT and KO hPSC were differentiated toward pancreatic cells using a well-established protocol[67] with modifications[68,69] (Supplementary Fig. 5D), which progressively directed cells toward adopting DE, pancreatic progenitor (PP), and then endocrine progenitor (EP) cell fate. We detected ETV1 protein in SOX17 + DE and PDX1 + PP cells (Fig. 5I). No significant difference in the formation of SOX17 + DE cells was observed between KO and WT

**Fig. 4 | Genes regulated by *ETV1/ETV4/ETV5* in hPSCs are associated with cell–cell and cell–matrix adhesion and PI3K/AKT pathway signaling.**
**A** Representative images of immunofluorescence staining of ITGA5 (red) protein in WT and KO hPSCs. DAPI marks the nuclei (blue). The zoomed inset highlights a selected area from the merged image. Scale bar = 50 μm. **B** Representative images of immunofluorescence staining of VCL (red) protein in WT and KO hPSCs. DAPI marks the nuclei (blue). The zoomed inset highlights a selected area from the merged image. Scale bar = 50 μm. **C** Western blot analysis of VCL protein expression in WT, KO, and tKO hPSCs. An antibody against GAPDH serves as a loading control. $N = 3$ biological replicates. **D** Representative images of immuno-fluorescence staining of CDH1 (green) and PXN (red) protein localization in WT, KO, and tKO hPSCs. CDH1 and PXN distribution within colonies is changed in KO and tKO compared to WT, and in KO compared to tKO. DAPI marks the nuclei (blue). The zoomed inset highlights a selected area from the merged image. Scale bar = 50 μm. **E** High-resolution SIM microscopy images (3D projections) of CDH1 (upper panel) and PXN (bottom panel) immunofluorescence staining in WT, KO, and tKO hPSC colonies. **F** Representative images of CDH1 (green) show disrupted colony integrity in tKO. $N = 3$ biological replicates. Scale bar = 100 μm. **G** High-resolution microscopy images of F-ACTIN (green) immunofluorescence staining show different actin cytoskeleton organization at the colony edges of WT and tKO hPSC. Scale bar = 20 nm. **H** Western blotting analysis of phospho-AKT (p-AKT), total AKT protein level in WT, KO, and tKO hPSCs. An antibody against GAPDH was used as a loading control. $N = 3$ biological replicates. **I** Representative bright-field images of WT and KO hPSCs (cells marked by a yellow mask) after 24 h treatment with PI3K/AKT pathway inhibitors (PI-103, Torin2) or an activator (insulin). Untreated and DMSO-treated WT and KO hPSCs served as controls. Scale bar = 400 μm.
**J** Quantification of crystal violet staining shows a dose-dependent decrease in the attachment of KO (green) after PI3K/AKT pathway inhibition by PI-103, compared to untreated (UT) and control (DMSO) KO hPSCs. $N = 4$ biological replicates. KO: UT vs. 5 nM, $p = 0.0003$, UT vs. 15 nM and UT vs. 30 nM, $p < 0.0001$; 30 nM KO vs. UT

WT, $p = 0.0422$. **K** Quantification of crystal violet staining shows a dose-dependent decrease in cell attachment following Torin2 treatment, compared to both untreated (UT) and DMSO-treated KO hPSCs. Treatment with 15 nM Torin2 resulted in a proportion of attached KO cells similar to untreated WT (yellow) hPSCs. Treatment with 200 nM Torin2 resulted in decreased attachment of KO compared to WT hPSCs. $N = 4$ biological replicates. KO: UT vs. 2 nM, $p = 0.0003$, UT vs. 15 nM, and UT vs. 200 nM, $p < 0.0001$; 15 nM KO vs. UT WT, $p =$ ns. **L** Quantification of crystal violet staining shows a dose-dependent increase of WT (yellow) attachment after insulin treatment, compared to untreated (UT) WT hPSCs. The 3 μg/ml insulin treatment resulted in a similar number of attached WT cells as in untreated (UT) KO hPSCs (WT: UT vs. 3 μg/ml, $p < 0.0001$; KO: UT vs. 3 μg/ml, $p =$ ns). The fraction of KO (green) hPSCs did not change following insulin treatment compared to UT KO hPSCs (all doses, $p =$ ns). No difference in the number of attached KO hPSCs was observed regardless of the insulin dose, indicating that ETV1 deletion may cause saturation of PI3K/AKT pathway activity. $N = 4$ biological replicates.
**M** Representative images of immunofluorescence of CDH1 (green) and PXN (red) proteins in Torin2 or insulin-treated WT and KO hPSCs. DAPI marks the nuclei in blue. Scale bar = 200 μm. **N** Quantification of fluorescence intensity (CDH1 + /DAPI) from immunofluorescence images of untreated (UT), DMSO-, Torin2- (TORIN2), and insulin-treated (INS) WT (yellow) and KO (green) hPSCs. $N = 3$ biological replicates. WT: UT vs. DMSO, $p =$ ns, UT vs. TORIN2, $p = 0.0016$, UT vs. INS, $p = 0.0447$; KO: UT vs. DMSO and UT vs. INS, $p =$ ns, UT vs. TORIN2, $p = 0.0001$. **N** Quantification of fluorescence intensity (PXN + /DAPI) from immunofluorescence images of untreated (UT), DMSO-, Torin2- (TORIN2) or insulin-treated (INS) WT (yellow) and KO (green) hPSCs. $N = 3$ biological replicates. WT: UT vs. DMSO, $p =$ ns, UT vs. TORIN2 and UT vs. INS, $p < 0.0001$; KO: UT vs. DMSO and UT vs. INS, $p =$ ns, UT vs. TORIN2, $p < 0.0001$. For plots (**J–L**, **N**, and **O**), a one-way ANOVA for multiple comparisons was used to determine the $p$-values shown on the graph. The data are presented as means ± SDs.

cells (Supplementary Fig. 5E, F). However, ETV1 KO cells exhibited reduced PDX1 expression and almost a complete lack of NKX6-1 protein at the PP stage, indicating a failure to form mature pancreatic progenitors (Fig. 5K, L, and Supplementary Fig. 5G). Consequently, KO PP cells did not progress to the EP cell fate, as we did not detect any CHGA+ endocrine cells in *ETV1* KO PPs (Fig. 5K and Supplementary Fig. 5G).

Intrigued by these findings, we conducted single-cell RNA-seq to delve into the transcriptome changes in *ETV1* KO PPs. We sequenced 5218 WT and 8810 KO cells at day 12 of differentiation (Fig. 6A). Using Seurat, we identified six molecularly distinct subpopulations: PP1, PP2, proliferating pancreatic progenitors (prolifPP), endocrine progenitors and endocrine cells (EP_EC), mesenchymal cells (mes), and liver progenitors (LP). All PP clusters (PP1, PP2, and prolifPP) were characterized by the expression of *PDX1*, *NKX6-1*, and *SOX9*. Additionally, prolifPP cluster could be distinguished from other PPs based on the increased expression of the well-known proliferation markers, such as DNA Topoisomerase II Alpha (*TOP2A*) or Marker of Proliferation Ki67 (*MKI67*) (Fig. 6B, C, and Supplementary Fig. 6A). Interestingly, PP1 and PP2 clusters differed in the expression of genes related to biological processes and pathways known-to-be involved in pancreatic development, such as NOTCH signaling, axon guidance, and EGF signaling. For example, PP1 cells showed increased expression of Semaphorins (*SEMA3A* and *SEMA6D*), while PP2 cells showed higher levels of Notch Receptor 2 (*NOTCH2*) and Slit Guidance Ligand 3 (*SLIT3*) (Fig. 6C, D, and Supplementary Fig. 6B). Transcripts uniquely expressed in the pancreatic endocrine lineage (EP_EC) included *NEUROG3*, *CHGA*, Neuronal Differentiation 1 (*NEUROD1*), Paired Box 4 (*PAX4*) or Aristaless Related Homeobox (*ARX*) (Fig. 6B, C, and Supplementary Fig. 6A). Moreover, the two clusters showing non-pancreatic fate, mes and LP, expressed distinct markers: mes cells expressed Vimentin (*VIM*) and various collagens (*COL3A1*, *COL5A*), while LP cells expressed Alpha Fetoprotein (*AFP*), Fibrinogen Beta Chain (*FGB*) and CF Transmembrane Conductance Regulator (*CFTR*) (Fig. 6B, C, and Supplementary Fig. 6A). Comparing cluster frequencies between KO and WT cells

showed a 30% and 40% decrease of EP_EC and PP1 clusters, respectively. Conversely, KO cells demonstrated a higher frequency of PP2 (by 80%), LP (by 300%), and mes (by 70%) clusters than WT PPs. The cell count of prolifPP was comparable between WT and KO cells (Fig. 6E). Concluding, changes in cell frequencies indicated disturbed PP development in KO compared to WT PPs.

Next, we focused on the PP2 cluster, which showed an 80% increase in KO compared to WT cells. Among upregulated genes in the PP2 cluster, we identified functional enrichment in the PI3K/AKT pathway, focal adhesion, axon guidance, and cell migration, indicating that ETV1 may control similar processes in hPSCs and during PP development (Fig. 6F). Among the upregulated transcripts in KO compared to WT PPs, we identified *VCL* and *COL4A* – two genes confirmed by ChIP-qPCR to be directly regulated in hPSCs by *ETV1* (Figs. 3L, 6G, H and Supplementary Fig. 6C). Furthermore, ATAC-seq analysis showed increased chromatin accessibility in the genomic regions encompassing *VCL* and *COL4A* in KO vs. WT PPs (Fig. 6I). Immunofluorescence staining confirmed higher protein levels of VCL and COL4A in KO compared to WT PPs (Fig. 6J–L). Altogether, these results suggest that ETV1-dependent adhesion changes and dysregulation of the PI3K/AKT signaling pathway affect PP maturation (Supplementary Fig. 6D), as shown by the discrepancies in early and late PP, and endocrine subpopulation frequencies in *ETV1* KO.

## Discussion

Biophysical parameters are important components of communication and interaction with the surrounding environment, influencing the reception and transmission of intra- and extracellular chemical signals or mechanical forces. Further, adhesion and its modifications affect cell shape, polarity, proliferation, and migration potential[70]. The impact of the microenvironment, including its biochemical and biophysical features, on cell behavior has become increasingly recognized in recent research.

While mechanobiology studies predominantly focus on outside-in signals, much less is known about the transcriptional mechanisms that

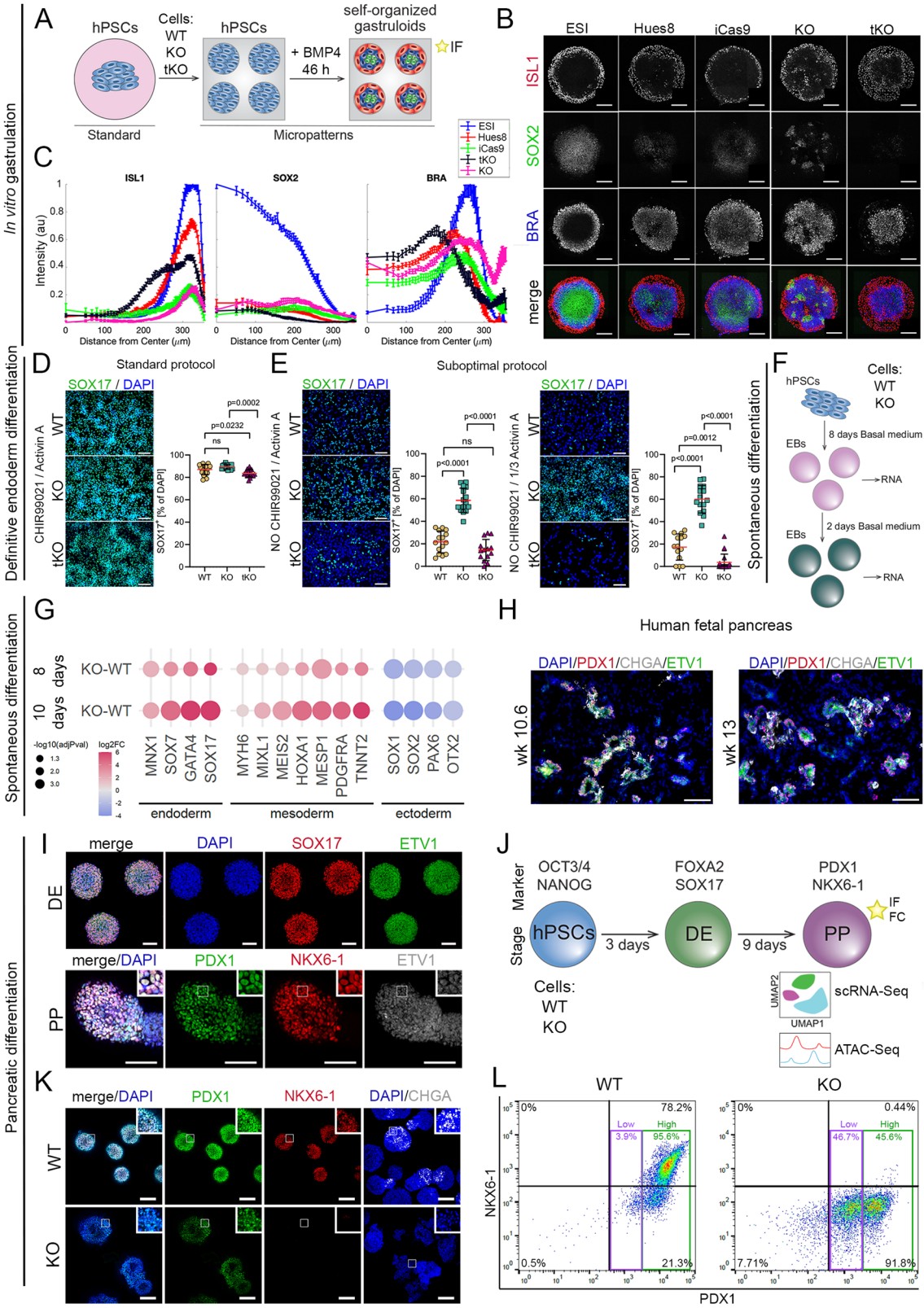

regulate intrinsic mechanical cues within the cell, such as adhesion to matrix and other cells. Here, we demonstrate that ETV1, alone and in conjunction with ETV4 and ETV5, controls hPSC adhesion strength to the other cells within the colony and ECM. Dysregulation of *ETV* expression in hPSCs leads to either enhanced attachment or detachment from the hPSC niche, in case of loss- or gain-of-function, respectively. It would be interesting to investigate cell tension in ETV1

KO or tKO hPSCs or PPs, as the increased adhesion properties likely affect cellular tension, further attuning mechanosignaling pathways.

Based on in silico analysis and ChIP-qPCR, ETVs directly repress the expression of cell–cell contact-associated genes such as *CDH1* or cell–matrix related transcripts including *VCL*, *PXN*, or *ITGA5*. Further, *ETVs* regulate the expression of PI3K/AKT pathway-related transcripts, such as *CDKN1A*, *CDKN1B* or *PDGFRB*. The PI3K/AKT signaling

**Fig. 5 | Deletion of the *ETV* genes affects germ-layer formation and pancreatic in vitro differentiation. A** Scheme illustrating the micropatterning approach to in vitro generation of the organized embryo-like structures (gastruloids). hPSCs (light blue) were plated on laminin 521-coated micropatterns and exposed for 46 h to BMP4 to induce self-organization and formation of gastruloids, containing extraembryonic-like cells (red), ectodermal cells (green) and the mesodermal and endodermal cells (dark blue). **B** Representative immunofluorescence staining of ISL1 (red), SOX2 (green), and BRA (blue) in gastruloids generated from ESI (hPSC control cell line), HUES8 (hPSC control cell line), iCas9 (WT hPSCs, parental control), KO and tKO hPSCs. Note, the disrupted germ layers organization in KO hPSCs (SOX2+ and BRA+), the overgrowth of extraembryonic-like cells (ISL1+), and the absence of SOX2+ cells in tKO, compared to control hPSC lines (ESI, HUES8 and iCas9). Scale bars = 200 μm. **C** Distribution from the center to the gastruloid edge of ISL1, SOX2, and BRA positive signals based on immunofluorescence staining on ESI (blue), HUES8 (red), iCas9 (green), tKO (black), and KO (pink) gastruloids. Data are presented as the mean ± SEM. $N = 3$ biological replicates. **D** WT, KO, and tKO differentiation to definitive endoderm using a standard protocol with full doses of Activin A and CHIR99021 (a WNT activator). $N = 3$ biological replicates. KO vs. WT, $p =$ ns, tKO vs. WT, $p = 0.0232$, tKO vs. KO, $p = 0.0002$. **E** WT, KO, and tKO differentiation to definitive endoderm using a suboptimal protocol with decreased concentration of Activin A, with or without CHIR99021. $N = 3$ biological replicates. KO vs. WT and tKO vs. KO, $p < 0.0001$, tKO vs. WT, $p =$ ns. **F** Scheme illustrating hPSC spontaneous differentiation into embryoid bodies (EBs). **G** Gene expression changes in KO EBs reveal altered differentiation potential. Dot plot shows log2FC in gene expression (qPCR) between KO and WT EBs at days 8 and 10 of spontaneous differentiation. The increased expression of endodermal and mesodermal markers and decreased expression of ectodermal markers were observed in KO at both timepoints. $N = 3$ biological replicates. The p-values were determined by two-sided student's t-test. **H** Representative immunofluorescence images of human fetal pancreas at weeks 10.6 and 13 of embryogenesis stained with PDX1 (red), CHGA (gray), and ETV1 (green). $N = 3$ tissue samples. Scale bar = 100 μm. **I** Representative immunofluorescence staining of ETV1 (green) in WT hPSC-derived definitive endoderm (DE, upper panel), and pancreatic progenitors (PP, bottom panel). SOX17 (red) marks DE, and PDX1 (green) and NKX6-1 (red) mark PPs. $N = 3$ biological replicates. Scale bar = 200 μm. **J** Scheme illustrating hPSC differentiation into pancreatic progenitors (PP). During differentiation, hPSCs pass through the following stages: definitive endoderm (DE) marked by FOXA2 and SOX17 expression, PP marked by PDX1 and NKX6-1 expression. At the PP stage (day 12) single-cell RNA sequencing and ATAC sequencing were performed on WT and KO samples. **K** Representative immunofluorescence staining of PDX1 (green), NKX6-1 (red), and CHGA (gray) proteins in WT and KO PPs. $N = 3$ biological replicates. Scale bar = 200 μm. **L** Representative flow cytometry analysis showing the absence of NKX6-1+KO PPs and decreased PDX1 protein levels in KO compared to WT PPs. Two distinct PDX1+ populations are observed in KO cells: low PDX1+ (~50% of cells) and high PDX1+ (~50% of cells). In contrast, WT PPs exhibit predominantly high PDX1 expression. For plots (**D**, **E**), a one-way ANOVA with Turkey's correction for multiple comparisons was used to determine the p-values shown on the graph. The data are presented as means ± SDs.

---

pathway regulates cell survival, proliferation, migration, and metabolism, during embryogenesis and in differentiated cells[71]. In hPSCs, PI3K/AKT pathway activation is required for self-renewal, ensuring active cell proliferation, as PI3K/AKT inhibition causes a slowdown of hPSC growth and upregulation of lineage-specific genes[72–74]. We observed upregulation of the PI3K/AKT pathway in KO and tKO hPSCs and experimentally linked this increased PI3K/AKT pathway activity with enhanced hPSC adhesion and expression of CDH1 and PXN. Although the highest increased confluency of ETV-deficient hPSCs and PI3K/AKT pathway activation might suggest increased proliferation, direct analysis of cell division markers (Ki67 and pHH3) did not show a corresponding increase in dividing cells. This advocates for the presence of an antagonistic mechanism. We propose that enhanced CDH1-mediated cell–cell contact in *ETVs* KO hPSCs increases cell packing density, which may in turn inhibit cell proliferation. The spatial organization of tightly packed hPSC colonies, held together by CDH1 interactions, could create local mechanical forces that control cell proliferation. This hypothesis aligns with other studies suggesting that mechanical interactions between cells, particularly through CDH1-mediated cell–cell adhesion, can influence cell proliferation[75–78].

To investigate potential changes in pluripotency caused by ETV1 KO, we performed a series of assays, including population RNA-seq of KO hPSCs, immunofluorescence staining, and flow cytometry. Our data revealed no obvious differences, suggesting that biased cell fate determination is not directly related to the pluripotency status of ETV1 KO. Instead, it may be associated with alterations in adhesion, and mechanosignaling mechanisms. We demonstrated aberrant cell fate specification and differentiation trajectories in KO and tKO hPSCs. Cell fate specification is a crucial process that begins during gastrulation, where pluripotent epiblast cells differentiate into the three germ layers. Gastrulation involves intricate cellular rearrangements to establish proper topological positioning. These rearrangements often coincide with lineage restriction and specification of germ layers, making adhesion alterations critical in this process[79]. To address whether initial cell fate is altered in hPSCs with higher adhesion caused by the deletion of *ETV* genes, we employed an in vitro micropatterned culture of gastrulation-like structures (gastruloids)[64]. We first noticed that the WT HUES8 hESC line used in this study exhibited a correct distribution of germ layers. However, it showed increased expression of the mesendoderm marker, BRA,

compared to another WT hESC line (ESI), likely due to inherent difference in differentiation propensity across distinct hPSC lines[66]. In KO and tKO gastruloids, we observed a disrupted germ-layer organization, suggesting that altered adhesion at the pluripotency stage biases initial cell fate decision resulting in a defective distribution of SOX2+ ectoderm, BRA+ mesendoderm, and ISL1+ extraembryonic-like cells. The increased cell adhesion in hPSCs after *ETV1* deletion may generate higher cell density and adhesion tension within 3D spheres, resulting in enhanced mesoderm and endoderm formation, as shown in in vitro spontaneous differentiation of KO cells. Cell adhesion molecules provide specificity of cell–cell or cell-ECM interactions, thus impacting signal transduction that regulates gene expression and spatial organization in developing embryos[80]. One plausible mechanism could involve differential accessibility to ligands, receptors, or signaling adapters in hPSCs with stronger cell–cell and cell–matrix attachment. Consistent with this hypothesis, we observed disrupted colony integrity in tKO colonies, with higher CDH1 levels at cell membranes but also gaps in cell–cell contact. This is consistent with work showing that epithelial integrity and receptor polarization can bias signaling toward the edge[81]. If these aspects are disrupted in ETV KOs, it could explain the altered germ-layer patterns. However, further research is required to gain a more precise understanding of how cell adhesion can potentially regulate cell fate specification.

We also investigated whether changes in adhesion influence cell fate decisions during directed in vitro differentiation, focusing on pancreatic specification as an example. In the murine pancreas, Etv1, Etv4, and Etv5 are expressed from E11.5 till birth. However, their expression patterns are distinct. *Etv1* is initially detected in bipotent pancreatic progenitors and later becomes restricted to endocrine cells, with high expression in α cells[42,82]. *Etv4* and *Etv5* are expressed in the distal epithelial tips, which are initially multipotent but later give rise to acinar cells[82,83]. In contrast, the human developing pancreas shows low levels of *ETV4* and *ETV5* expression, primarily in a few mesenchymal cells[84]. Using single-cell RNA-seq datasets of human fetal pancreas (8-20 weeks post-conception), we determined that ETV1 mRNA is predominantly expressed in all endocrine cell types, with the highest levels detected in α and δ cells, and barely detectable expression in the mesenchymal compartment[84]. During hPSC differentiation toward β cells, ETV4 and ETV5 expression ceases after hPSC and DE stages[85]. Conversely, ETV1 shows high expression during

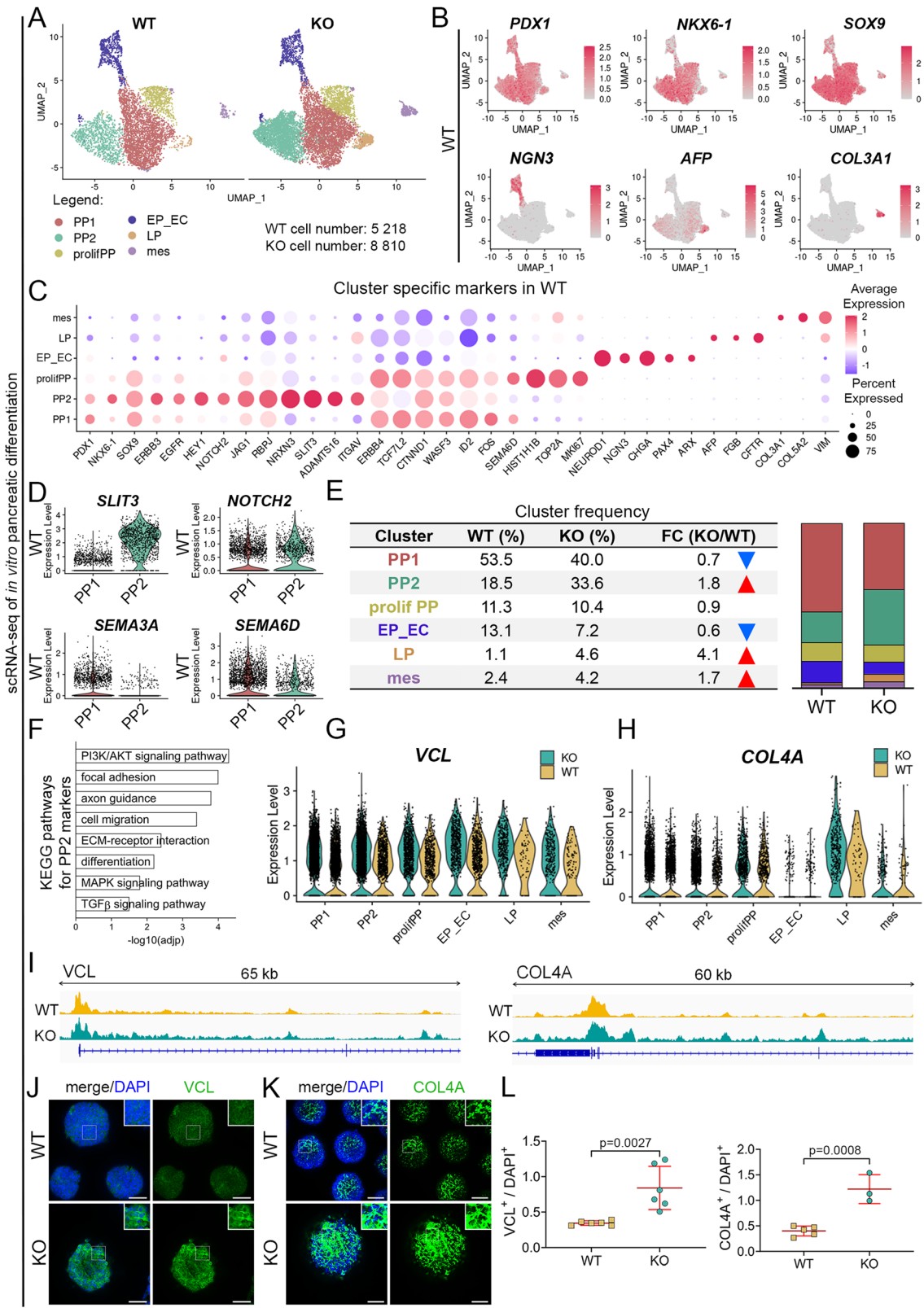

pluripotency, decreases at intermediate stages, and then increases again in PPs and endocrine cells, particularly α cells[85]. This is consistent with ETV1 being a well-established marker of α cells in the pancreas. Additionally, *Etv1* KO mice exhibit decreased islet size, indicating a role of *Etv1* in morphogenesis of the pancreatic endocrine compartment[82].

Our study reveals that ETV1 functions much earlier than previously thought, acting upstream of human endocrine cell formation. *ETV1* KO resulted in altered PP maturation and decreased EP formation. This highlights a crucial role for ETV1 in regulating the transition of multipotent pancreatic progenitors into EPs and raises questions about progenitor specification and arrangement during pancreas

**Fig. 6 | *ETV1* deletion inhibits pancreatic progenitor in vitro specification into endocrine cells. A** Single-cell sequencing reveals distinct cell populations within WT and KO samples. UMAP plots visualize the distribution of 5218 and 8810 WT and KO cells, respectively. Clusters are identified as follows: PP1 (red) – pancreatic progenitors 1, PP2 (green) – pancreatic progenitors 2, prolifPP (yellow) – pro-liferating pancreatic progenitors, EP_EC (blue) – endocrine progenitors and endo-crine cells, LP (orange) – liver progenitors, mes (purple) – mesenchyme cells. **B** Feature plots confirm cluster identities based on marker expression: *PDX1, NKX6-1,* and *SOX9* for PP1, PP2, and prolifPP; *NEUROG3* for endocrine progenitors and endocrine cells; *AFP* for liver progenitors, *COL3A1* for mesenchyme cells. **C** Dot plot represents the average expression and percentage of cells expressing markers associated with a specific cell type or cluster. *PDX1, NKX6-1,* and *SOX9* are highly expressed in PP1, PP2, and prolifPP clusters. PP1 and prolifPP clusters show increased expression of genes related to cell adhesion, while the prolifPP cluster shows an elevated expression of proliferation markers (*HIST1H1B, TOP2A,* and *MKI67*). The PP2 cluster demonstrates the expression of genes connected to the Notch signaling pathway, axon guidance, and cell migration. The EP_EC cluster expresses *NEUROD1, NEUROG3, CHGA, PAX4* and *ARX*. Liver progenitors (LP) show increased expression of genes related to liver function, including *AFP, FGB,* and *CFTR*. Enhanced expression of *COL3A1, COL5A2,* and *VIM* is observed in mesenchyme subpopulation (mes). **D** Violin plots show the differential expression of *SLIT3, NOTCH2, SEMA3A,* and *SEMA6D* between WT pancreatic progenitors 1 (PP1) and pancreatic progenitors 2 (PP2) clusters. The expression of *SLIT3* and *NOTCH2* is higher in the PP2 cluster compared to the PP1 cluster, while the expression of *SEMA3A* and *SEMA6D* is increased in the PP1 compared to the PP2 cluster. **E** Table and bar plot show the cluster frequencies in WT and KO cells. In KO a decreased

frequency is observed for the pancreatic progenitors 1 (PP1, red), and endocrine progenitors and endocrine cells (EP_EC, blue) clusters. Conversely, PP2 (PP2 green), liver progenitors (LP, orange) and mesenchyme (mes, purple) clusters show increased frequencies in KO cells compared to WT cells. The frequency of pro-liferating pancreatic progenitors (prolifPP, yellow) is similar for both WT and KO cells. **F** Bar plot shows significantly enriched functional terms (adjusted p-value ≤ 0.05, KEGG pathway database) related to the PI3K/AKT signaling pathway, cell adhesion, and cell migration among genes upregulated in pancreatic progenitor 2 (PP2) cluster. **G** Violin plot demonstrates increased expression of *VCL* in KO (green) compared to WT (yellow) cells. PP1 – pancreatic progenitors 1, PP2 – pancreatic progenitors 2, prolifPP – proliferating pancreatic progenitors, EP_EC – endocrine progenitors and endocrine cells, LP (orange) – liver progenitors, mes – mesench-yme cells. **H** Violin plot demonstrates increased expression of *COL4A* in KO (green) compared to WT (yellow) cells. PP1 – pancreatic progenitors 1, PP2 – pancreatic progenitors 2, prolifPP – proliferating pancreatic progenitors, EP_EC – endocrine progenitors and endocrine cells, LP (orange) – liver progenitors, mes – mesench-yme cells. **I** ATAC-seq tracks demonstrate *VCL* (left) and *COL4A* (right) loci in WT (yellow) and KO (green) PPs. The peaks represent normalized and combined bio-logical replicates (*N* = 2). **J** Representative images of VCL (green) immuno-fluorescence staining in KO and WT PP spheres. DAPI marks the nuclei (blue). Scale bar = 200 μm. **K** Representative images of COL4A (green) immunofluorescence staining in WT and KO PP spheres. DAPI marks the nuclei (blue). Scale bar = 200 μm. **K** Quantification of fluorescence intensity from immunofluorescence staining in WT (yellow) and KO (green) PPs (day 12). *N* = 3 biological replicates. Two-sided student's *t*-test was used to determine the *p*-values shown on the graph. Data are presented as the mean ± SDs.

development. In pancreas, both exocrine and endocrine cells origi-nate from the same progenitor pool, progressively differentiating after bud formation. Endocrine cells emerge from a central core domain, while acinar cells arise from the peripheral tip cells[86,87]. One hypothesis on the mechanisms of establishing these distinct domains is that cells with similar adhesion molecules cluster toge-ther to form a more energetically favorable state[88,89]. Our findings support this, demonstrating that increased adhesion of early pan-creatic progenitors hinders their differentiation into the endocrine cells. Single-cell RNA-seq analysis of WT and *ETV1* KO PPs suggest two possible outcomes for *ETV1*-deficient PPs: (1) arrested development at the PP2 stage, indicating blocked endocrine lineage progression, or (2) differentiation toward alternative lineages, evidenced by the acquisition of CPA2 or AFP expression. Interestingly, single-cell RNA-seq analysis also revealed an increase in hepatocyte-like and mesenchyme-like cells in ETV1 KO cultures, even under conditions promoting pancreatic specification. Noteworthy, both KO and tKO cells, both at the hPSCs and PP stage, exhibit increased resistance to dissociation, requiring more robust enzymatic treatment to obtain a single-cell state.

Finally, we observed *ETV*-dependent changes in adhesion, reflected by increased expression of CDH1 and PXN, VCL, ITGA5, and various collagens, such as COL4A6, COL13A1, and COL7A1, as well as altered F-actin distribution, suggesting potential alterations in cell behaviors like migration and epithelial-to-mesenchymal transition (EMT). Interestingly, ETV1 is crucial for neural crest cell development in *Xenopus laevis*[48,90], where EMT is essential for proper differentia-tion. Additionally, the dynamic formation of defined actin fibers reflects cell's ability to adapt to the microenvironment changes or participate in processes, like cell migration[91]. This highlights the potential link between ETV-regulated biophysical features and cell fate specification.

Our findings demonstrate that manipulating biophysical proper-ties, such as cell–cell and cell-ECM adhesion, and cytoskeleton, which contribute to cell mechanosignaling, can influence cell fate specifica-tion in hPSCs. This suggests a combined approach, targeting cell adhesion and cell fate signals, may be necessary to optimize in vitro hPSC differentiation strategies for regenerative medicine and embry-ogenesis research.

## Methods
### hPSC culture
All experiments with human embryonic stem cells were performed in accordance with relevant guidelines and regulations of the Local Bioethics Committee of Medical University in Poznan, Poland. hPSCs with doxycycline-inducible Cas9 expression (HUES8-iCas9) were established in Dr. Huangfu laboratory (MSCK, NYC, USA), through targeting into AAVS1 safe harbor locus, the inducible Cas9 cassette using TALEN approach[62]. HUES8-iCas9 hPSCs were cultured in Stem-Flex (Thermo Fisher Scientific, Netherlands) medium on Geltrex-coated (Thermo Fisher Scientific, Netherlands) plates at 37 °C, under an atmosphere containing 5% CO$_2$. Cells were passaged every 3–4 days using PBS-EDTA. Cell lines were routinely tested for mycoplasma with PCR assay and found negative.

### Deletion of *ETV1* (KO) and *ETV1*, *ETV4* and *ETV5* genes (tKO) in hPSCs
Different sgRNAs targeting the sequence of exon 4 for *ETV1*, exon 2 and 3 for *ETV4*, and exon 3 and 4 for *ETV5* were designed using the Benchling software. The sgRNA sequences are listed in Supplementary Table 1. The sgRNAs were produced in-house using previously pub-lished protocol[92]. Briefly, the PCR template constituted a 120-nucleotide single-stranded DNA including a T7 promoter, sgRNA tar-get-specific, and constant sgRNA sequences. The PCR product was used for in vitro transcription to generate sgRNAs for lipofection. The sgRNAs targeting exon 9 for *ETV1* were designed and produced by Synthego (USA). Doxycycline-treated HUES8-iCas9 cells were dis-persed to single ones using TrypLE Select (Thermo Fisher Scientific, Netherlands), counted, and seeded in E8 medium (Thermo Fisher Scientific, Netherlands) with 10 μM Y-27632 (Peprotech, USA) into Geltrex-coated 24-well plate at the density of $1.5 \times 10^5$ cells/well. The cells were reversely transfected using Lipofectamine RNAiMax (Thermo Fisher Scientific, Netherlands) and cultured for 2 days. Subsequently, the cells were plated at single-cell density onto Geltrex-coated 6 mm dishes, cultured for the following 7 days, and single colonies were transferred into a 96-well plate. PCR on genomic DNA isolated from the clones with the primers flanking the CRISPR-edited region, electrophoresis of amplified DNA, and Sanger sequencing allowed to identify the clones with homozygous

KO mutation in *ETV1*, *ETV4*, or *ETV5* gene. In the case of tKO, *ETV1* KO was used as a background cell line. CRISPR-edition efficiency was calculated using the bioinformatic tool Inference of CRISPR Edits[93], based on the DNA sequencing comparison of edited and control samples.

## RNA sequencing

We used 500–1000 ng of total RNA isolated from hPSCs to prepare libraries with the TruSeq RNA Library Prep Kit v2 according to the manufacturer's protocol. Libraries were prepared in duplicate. Libraries were quantified with a Qubit fluorometer (TFS), and their quality was assessed with the Agilent High Sensitivity DNA kit (Agilent Technologies, USA). Libraries were sequenced with an Illumina HiScanSQ sequencer. RNA-seq raw paired-end reads were trimmed with fastp. Trimmed reads were aligned with the Ensembl GRCh38 human genome, with STAR v2.7, and raw counts were obtained with featureCounts v1.6.3. RNA-seq raw sequence reads were analyzed with the iDEP9.4 (integrated Differential Expression and Pathway analysis) interactive platform[94]. Genes differentially expressed (DEGs) between KO and WT, tKO and WT, or tKO and KO were identified and normalized with the DESeq2 package. To normalize data for technical variation between RNA-seq experiments the RUVSeq R package was used. The RNA-seq data have been deposited in the NCBI GEO database under accession number GSE227794. The analyzed datasets can also be found in the Supplementary Data 1. The DEGs significantly (*p*-value ≤ 0.05) upregulated with a log2 FC ≥ 0.5 or downregulated with a log2 FC ≤ −0.5 in any of the sequenced samples were selected for further analysis. The set of DEGs in KO vs. WT and tKO vs. WT were compared to each other in the Venn diagram analysis. Volcano plots were plotted with GraphPad Prism v8.4.2 for Windows (GraphPad Software, Boston, Massachusetts USA) for genes with normalized reads, with −log10 adjusted *p*-value (−log10 adjp) values on the *y* axis and log2 FC values on the *x* axis. Genes with −log10 adjp ≥ 2 and log2 FC ≥ 0.5 or log2 FC ≤ −0.5 were considered as significant. The GeneCodis interactive platform[95] was used for the functional enrichment analysis to identify pathways and processes among DEGs based on the Kyoto Encyclopedia of Genes and Genomes (KEGG), Biological Processes (BP), and Wiki Pathways databases. The significance cutoff value (*p*-value < 0.05) was utilized in functional enrichment analysis. Heatmaps of representative genes from selected KEGGs, BPs, and WikiPathways were plotted with GraphPad Prism 8 software. The DNA binding prediction for ETVs among DEG promotors was performed in silico with the Pscan tool[96]. ETV binding motif matrix derived from a curated database of transcription factors[97] was searched against DEG promoter regions set as −950/+50 bp from TSS (transcription start site). DEGs with the z-score >0.8 were selected as those with the potential ETV's binding.

## Single-cell RNA sequencing

Pancreatic progenitor spheres, WT and KO, were dissociated into single cells using 1% trypsin in PBS at 37 °C. Libraries for single-cell RNA sequencing were conducted with 10x Genomics kit according to the manufacturer's protocol. Single-cell transcriptome was marked with UMI barcode using a droplet-based method in Chromium (10x Genomics, USA). For quality and quantity assessment of cDNA libraries, Agilent TapeStation was used with High Sensitivity D1000 ScreenTape (Agilent Technologies, USA). Libraries were paired-end sequenced with a depth of 40,000 reads per cell. 5218 and 8810 cells were sequenced for WT and KO pancreatic progenitors respectively. Bioinformatic analyses were conducted with scripts for CellRanger (10x Genomics, USA), including WT and KO data aggregation with "cellranger agrr" algorithm. The scRNA-seq data have been deposited in the NCBI GEO database under accession number GSE227794. Single-cell data clustering and visualization with the UMAP algorithm, analysis of specific markers or DEGs for each cluster (using Wilcoxon

Rank Sum test), and violin and feature plots with gene expression patterns within the clusters were performed in R Studio using Seurat Package v4.3.0. Enriched pathways and processes among markers for each cluster and DEGs were identified via the Kyoto Encyclopedia of Genes and Genomes (KEGG), Biological Processes (BP), or Wiki Pathways databases. The functional enrichment analysis was performed in the GeneCodis interactive platform[95]. The significance cutoff value (adj p-value < 0.05) was utilized in functional enrichment analysis.

## Micropatterning – gastruloid model

ESI017 hPSCs were obtained from ESI BIO and used as a control as they have shown robust patterning in our previous work[65,98,99]. All lines for micropatterning were grown in mTeSR-1 or mTeSR-Plus medium (STEMCELL Technologies, Canada) in tissue culture dishes coated with Geltrex (Gibco, USA 1:500 in DMEM/F12). Cells were passaged using Dispase (STEMCELL Technologies) for regular maintenance and with Accutase (Corning, USA) for micropatterning experiments. The pluripotency of all cell lines was confirmed via immunostaining for markers OCT3/4, SOX2, and NANOG. All cells were routinely tested for mycoplasma contamination and found negative.

Micropatterning experiments were performed in 96-well plates obtained from CYTOO. The surface was coated with 5 µg/ml Laminin 521 in PBS with Ca/Mg for at least 2 h and then cells were seeded in mTeSR-1 with ROCK inhibitor Y27672 (10 µM, MedChemExpress, USA) using the protocol described before[100]. We noticed that following seeding ESI017 cultures contained more cells on the micropatterns than the other lines and therefore seeded $2 \times 10^5$ ESI017 cells or $2.5 \times 10^5$ cells for all other lines. Three hours after seeding, media was replaced with mTeSR-1 with 50 ng/ml BMP4 (R&D Systems) and without ROCK inhibitor. Cells were incubated for 46 h, then fixed for 30 min with 4% PFA and stained.

## Spontaneous hPSC differentiation

The hPSC colonies were dissociated into single cells with TrypLE Select (Thermo Fisher Scientific, Netherlands). Cells were plated in E8 medium (Thermo Fisher Scientific, Netherlands) supplemented with 10 µM Y-27632 (Peprotech, USA) at the initial density of $2.5 \times 10^6$ cells/well on non-treated low-adhesion 6-well plate (Eppendorf, USA). Plates were placed on orbital shakers at 110 rpm to increase the uniformity of EBs. EBs were cultured in DMEM/F12 advanced medium (Thermo Fisher Scientific, Netherlands) with the addition of 1% Pen/Strep (Thermo Fisher Scientific, Netherlands), 1% Glutamax (Thermo Fisher Scientific, Netherlands), 1% MEM Non-Essential Amino Acids Solution (Thermo Fisher Scientific, Netherlands) and 2% heat-inactivated FBS (Merck, USA). Following the first 2 days of culture, 2% heat-inactivated FBS in the medium was changed to 10% heat-inactivated FBS. The medium was changed daily. After 8 and 10 days EBs were harvested for mRNA isolation. Gene expression levels of known makers for three germ layers were analyzed by the qPCR approach.

## hPSC pancreatic differentiation

For in vitro pancreatic differentiation, we used previously established protocol[67] with modifications[68,69]. hPSCs were dispersed into single-cell suspensions with TrypLE Select (Thermo Fisher Scientific, Netherlands) and plated at a density of $3.5 \times 10^5$ cells/cm² in E8 (Thermo Fisher Scientific, Netherlands) medium supplemented with 10 µM Y-27632 (Peprotech, USA) on low-adhesion non-treated 6-well plates (Eppendorf, Germany) on orbital shaker at 37 °C, 110 rpm and under an atmosphere containing 5% $CO_2$. On subsequent days, the medium was changed to fresh E8 medium without Y-27632. The next day the spheres were washed in DMEM/F12 medium (Corning, USA) and the medium was changed, as follows:

Basal medium: S1 – MCDB131 (Corning, USA), 1% Glutamax (Thermo Fisher Scientific, Netherlands) + 1% Pen/Strep (Thermo Fisher Scientific, Netherlands), 2.44 mM glucose (Merck Millipore, USA), 29 mM sodium bicarbonate (Merck Millipore, USA), 2% BSA (Capricorn Scientific, USA), 2 µl/100 ml ITS-X (Thermo Fisher Scientific, Netherlands). S2 – MCDB131 (Corning, USA), 1% Glutamax (Thermo Fisher Scientific, Netherlands) + 1% Pen/Strep (Thermo Fisher Scientific, Netherlands), 2.44 mM glucose (Merck Millipore, USA), 13 mM sodium bicarbonate (Merck Millipore, USA), 2% BSA (Capricorn Scientific, USA), 2 µl/100 ml ITS-X (Thermo Fisher Scientific, Netherlands). S3 – MCDB131 (Corning, USA), 1% Glutamax (Thermo Fisher Scientific, Netherlands) + 1% Pen/Strep (Thermo Fisher Scientific, Netherlands), 2.44 mM glucose (Merck Millipore, USA), 13 mM sodium bicarbonate (Merck Millipore, USA), 2% BSA (Capricorn Scientific, USA), 500 µl/100 ml ITS-X (Thermo Fisher Scientific, Netherlands).

Day 1: S1 + 3 µM CHIR99021 (Peprotech, USA) + 100 ng/mL Activin A (Peprotech, USA) + 250 µM vitamin C (Sigma Aldrich, USA); Days 2–3: S1 + Activin A; Days 4–6: S2 + 50 ng/ml KGF (Peprotech, USA) + 1.25 µM IWP2 (Selleckchem, USA) + 250 µM vitamin C; Day 7: S3 + 50 ng/ml KGF + 2 µM retinoid acid (Peprotech, USA) + 500 nM PdBu (Tocris, United Kingdom) + 250 nM SANT-1 (Tocris, United Kingdom) + 200 nM LDN193189 (Peprotech, USA) + 250 µM vitamin C + 10 µM Y-27632; Days 8-12: S3 + 5 ng/ml Activin A + 50 ng/ml KGF + 100 nM retinoid acid + 250 nM SANT-1 + 1.25 µM IWP2 + 100 ng/ml EGF (Peprotech, USA) + 10 mM nicotinamide (Sigma Aldrich, USA) + 250 µM vitamin C.

### Live-cell imaging and analysis
The IncuCyte live-cell imager (Sartorius, Germany) was used to track living cells. hPSCs were seeded at cell density $1.5 \times 10^4/cm^2$ on 24-well Geltrex-coated plates and cultured in IncuCyte for a maximum of 4 days at 37 °C under an atmosphere containing 5% $CO_2$. For live-cells quantitative analysis, 50 nM SiR DNA and 100 nM Verapamil (Tebubio, France) were added to the cells along with the passage. Photomicrographs were taken every 2 h and the confluency of cultures was measured in real time with the IncuCyte Base Analysis Software (Sartorius, Germany). Cell growth was monitored by analyzing the change in the area of an image occupied by the cells (confluency in %) over time.

### Treatment with cell signaling perturbants
The KO and WT hPSCs were used to seed on Geltrex-coated plates at a density of $1.5 \times 10^4$ cells/cm². The cells were treated for 24 h with modulators of PI3K/AKT pathway: PI-103 (AdooQ, USA) at final concentrations of 5, 15, or 30 nM; Torin2 (STI, USA) at a final concentration of 2.1, 15, or 200 nM; insulin (NovoNordisk, Denmark) at a final concentration of 0.03, 0.3 or 3 µg/ml. DMSO treatment (1:1000) was used as a vehicle control.

### Precoating of culture substrates
For the adhesion assay, cell culture treated plates (BD Falcon, USA) were coated with Laminin V (Thermo Fisher Scientific), Geltrex (Thermo Fisher Scientific, Netherlands), Fibronectin (Merk, USA), or rhVTN-N (20–398 aa of Vitronectin; Wako, USA), all at the concentration of 1 µg/cm² for 1 h at 37 °C, in $CO_2$ incubator just prior to cell seeding. Dulbecco's phosphate-buffered saline (DPBS, Thermo Fisher Scientific, Netherlands) or DMEM/F12 (Thermo Fisher Scientific, Netherlands) was used to dissolve the culture substrates. The solution volume for coating was standardized to 100 µl/cm².

### Crystal violet analysis
The KO, tKO, and WT hPSCs were seeded at cell density $1.5 \times 10^4/cm^2$ on different ECM coatings (Laminin, Fibronectin, Vitronectin, or Geltrex) and culturing for 24 h before crystal violet staining. For staining, the cells were fixed with 4% paraformaldehyde/PBS for 15 min at room temperature and washed with PBS. Next, the cells were stained for 30 min with freshly prepared 0.1% crystal violet in $H_2O$. To lyse cells for dye release, the hPSCs were incubated with 2% SDS for 30 min. The absorbance was measured at 550 nm by TECAN Sparc (Tecan, Switzerland). Measurements were normalized to the absorbance value from the well without cells, coated only with an appropriate matrix.

### Immunofluorescence staining
Human pancreas sections were processed at Baylor College of Medicine, Houston, TX (USA) with Institutional Review Board guidelines at Baylor College of Medicine–H-3097 approval granted to Malgorzata Borowiak. Donor identities were encrypted, and the data were analyzed anonymously. The informed consent was obtained from all human research participants. Human 10.6- and 13-week fetal pancreas samples were fixed in 4% paraformaldehyde/PBS for 4 h, washed with PBS, soaked in 30% sucrose, and embedded in TissueTek. Sections (12 µm-thick) were cut onto Superfrost Plus-coated glass slides and stored at −80 °C.

Cell monolayers were fixed with 4% paraformaldehyde/PBS for 15 min at room temperature and washed with PBS. Fixed cells were permeabilized by incubation with 0.5% Triton X-100 (BioShop, Canada) in PBS for 15 min and blocked by incubation for 45 min with 5% normal donkey serum (NDS, Jackson ImmunoResearch, UK) 0.1% Tween-20 (BioShop, Canada) in PBS. The samples were then incubated overnight at 4 °C with primary antibodies diluted in 5% NDS. The next day, cells were washed three times in 0.1% Tween-20 in PBS and proceeded for incubation with secondary antibodies conjugated with Alexa Fluor 488, TRITC, or Alexa Fluor 647 diluted 1:400 in 5% NDS, for 2 h at room temperature. The excess of the secondary antibody was removed by two washes in 0.1% Tween-20 in PBS, and the samples were then incubated with DAPI (Sigma Aldrich, USA) as a counterstain.

3D spheres were fixed with 4% paraformaldehyde/PBS for 45 min at 4 °C and washed once with 0.1% Tween-20 in PBS and suspended in spheroid wash buffer (SWB: 2% BSA, 0.1% Triton X-100, 0.025% SDS, 1× PBS). Fixed spheres were blocked in SWB for 45 min at 4 °C and incubated overnight with primary antibody at 4 °C on an orbital shaker. The next day the spheres were washed 3 times with SWB solution for 1 h each at 4 °C, and incubated with secondary antibodies conjugated with Alexa Fluor 488, TRITC or Alexa Fluor 647 diluted 1:400 and DAPI (1:10,000) in SWB, overnight at 4 °C on orbital shaker. On the next day, the excess of the secondary antibody was removed by 2 washes in 0.1% Tween-20 in PBS, first rapid, second for 1 h at 4 °C on an orbital shaker. For imaging, cells were kept in 1× PBS on a high content imaging glass-bottom 96-well plate (Corning, USA).

Primary and secondary antibodies used in the study are listed in Supplementary Table 2.

### Microscopy Imaging
Images were obtained with an epifluorescence, confocal, or super-resolution microscope. Bright-field images were taken with a Leica DM IL-Led (Leica, Germany) microscope with N Plan Fluor 4×/0.12, N Plan Fluor 10×/0.30, N Plan Fluor 20×/0.40 and N Plan Fluor 40×/0.60 lenses and a JENOPTIK GRYPHAX series ProgRes camera (JENOPTIK, Germany). Fluorescence images were obtained with a confocal microscope Nikon A1Rsi (Nikon, Germany) microscope with Plan Fluor 4×/0.13, Plan Apo 10×/0.45 DIC N1, Plan Apo VC 20×/0.75 DIC N2, Apo 40×/1.25 WI λS DIC N2, and Plan Apo VC 60×/1.4 Oil DIC N2 lenses and with Nikon NIS Elements AR 5.21.01 64-bit software (Nikon, Germany). Super-resolution microscopy was performed with a ZEISS Elyra 7 with Lattice SIM (Zeiss, Germany) microscope with Plan-Apochromat 40x/1.4 Oil DIC M27 (Immersol 518F – 23 °C/30 °C) and Plan-Apochromat 63×/1.40 Oil DIC f/ELYRA (Immersol 518F – 23 °C/30 °C) lenses and with ZEN Black 3.1 software (for imaging) and ZEN Blue 3.1 software for data analysis (Zeiss, Germany).

## Subcellular fractionation

Subcellular fractions were obtained following a previously published protocol[101], with modifications. A total of $1 \times 10^6$ the WT hPSCs were lysed with NETN buffer (20 mM Tris–HCl), pH = 8.0 (BioShop, Canada), 100 mM NaCl (BioShop, Canada), 1 mM EDTA (Sigma Aldrich, USA) and 0.5% Nonidet P-40 (Biosciences, USA), containing 1 μl Halt Protease Inhibitor Cocktail (Thermo Scientific, USA) for 30 min at 4 °C. Lysates were saved as the soluble fraction by centrifugation at $16,000 \times g$ for 30 min at 4 °C. Pellets were digested in 100 μl NEB buffer containing 3 μl Micrococcal Nuclease and 5 μl CaCl₂, with the addition of 1 μl Halt Protease Inhibitor Cocktail (all from Subcellular Protein Fractionation Kit for Cultured Cells, Thermo Scientific, USA), for 5 min at 37 °C. The chromatin-bound proteins were cleared at $16,000 \times g$ for 5 min at 4 °C to remove debris. 10 μg of proteins from both fractions were used for the ETV1 and pHH3 western blotting as described below. Protein from total cell lysate was used as a positive control.

## Western blot

Protein extracts from cells were prepared in Radio Immunoprecipitation Assay (RIPA) buffer (50 mM Tris–HCl), pH = 8.0 (BioShop, Canada), 150 mM NaCl (BioShop, Canada), 1% Nonidet-40 (BioShop, Canada), 0.5% sodium deoxycholate (BioShop, Canada), 0.1% SDS (BioShop, Canada), 1% protease inhibitor cocktail (Sigma Aldrich, USA), 1% EDTA (Sigma Aldrich, USA), 0.1% PMSF (Sigma Aldrich, USA), and stored at −80 °C. Protein concentration was determined with a BCA Protein Assay Kit (Thermo Fisher Scientific, Netherlands). 30 μg of protein in Bolt LDS buffer (Thermo Fisher Scientific, Netherlands) was pretreated at 70 °C for 10 min and loaded onto a 4–12% Bis-Tris Plus Gel (Thermo Fisher Scientific, Netherlands). Electrophoresis was performed at 165 V for the first 10 min, and 200 V for an additional 30 min, with a constant current of 100 amps. The resulting bands were transferred onto the PVDF (Thermo Fisher Scientific, Netherlands) membrane for 10 min in BioRad Trans-Blot Turbo Transfer System. The membranes were then blocked with 0.125% non-fat dry milk or 1–3% BSA in TBS-Tween-20 and incubated with primary antibodies overnight and next day with HRP-conjugated secondary antibodies as indicated in Supplementary Table 3. Antibody-antigen complexes were visualized by enhanced chemiluminescence (ECL) with the Luminata Forte HRP Substrate (Merck Millipore, USA) and detected with the G:Box System (Syngene, India). Western blots were quantified in ImageJ.

## Flow cytometry

Cells were dissociated with TrypLE Select (Thermo Fisher Scientific, Netherlands) to obtain a single-cell suspension. Cells were fixed by incubation with 4% paraformaldehyde in PBS with 0.1% saponin for 45 min at 4 °C and washed with 0.1% saponin – 1% BSA – PBS (SBP). The resuspended cells were incubated overnight at 4 °C on the roller with primary antibodies diluted in SBP. The following day, the cells were washed twice with SBP and incubated with secondary antibodies conjugated with TRITC, Alexa Fluor 488, or Alexa Fluor 647 (Jackson ImmunoResearch, UK) diluted between 1:400 and 1:600 in SBP for 1 h at room temperature. The excess secondary antibodies were removed by washing once with PBS and the cells, resuspended in PBS, were used directly for flow cytometry analysis. Flow cytometry data were acquired with a CytoFLEX Flow (Beckman Coulter, USA), or Guava EasyCyte HT (Millipore, USA), or NovoCyte Flow Cytometer (Agilent Technologies, USA). Flow cytometry data analysis, including gating, quantification, and the generation of density plots/histograms, was performed with FlowJo Software v10 (BD, USA) or NovoExpress Software v1.6.2 (Agilent Technologies, USA). The primary and secondary antibodies used in the study are listed in Supplementary Table 4.

## Physical cytometer

hPSC 3D spheres were dispersed into single-cell suspensions with TrypLE Select (Thermo Fisher Scientific, Netherlands) and plated at a density of $3.5 \times 10^5$ cells/cm² in E8 (Thermo Fisher Scientific, Netherlands) medium supplemented with 10 μM Y-27632 (Peprotech, USA) on low-adhesion non-treated 6-well plates (Eppendorf, Germany) on orbital shake at 37 °C, under an atmosphere containing 5% CO₂. The next day, spheres were fixed with 4% paraformaldehyde/PBS for 45 min at 4 °C, washed once with PBS, and suspended in 1% BSA in PBS. For the physical cytometer, fixed spheres were suspended in 7 ml of W8 Analysis Solution and analyzed in W8 Physical Cytometer (Cell Dynamics, Italy).

## Chromatin immunoprecipitation

Chromatin immunoprecipitation was performed based on the protocol from Schmidt et al. [102] on WT hESCs. Briefly, $1–3 \times 10^7$ cells were fixed with 1% PFA in 50 mM HEPES-KOH pH 7.4, 100 mM NaCl, 1 mM EDTA, and 0.5 mM EGTA for 10 min at room temperature. Glycine quenched the fixation, followed by sequential washes with PBS and a series of lysis buffers (L1, L2, and L3) supplemented with protease inhibitors. Subsequently, cells were sonicated using Bioruptor 300 sonicator (Diagenode, Belgium) to achieve chromatin shearing within the desired range (200–500 bp), as verified by agarose gel electrophoresis. To isolate sheared chromatin fragments, 10% Triton X-100 was added, followed by centrifugation to remove cellular debris. The supernatant was immunoprecipitated overnight at 4 °C with either anti-ETV1 antibody (20 μg; Thermo Fisher Scientific, cat. number: PA5-67447) or control IgG antibody (1 μg; Thermo Fisher Scientific, cat. number: 02-6102) using magnetic protein A/G beads. After washes with RIPA buffer and TBS buffer, immunoprecipitated DNA-protein complexes were eluted with proteinase K and subjected to reversal of crosslinking and proteinase K digestion. Purified DNA was then analyzed by qPCR for enrichment of target promoter regions associated with ETV1 binding.

## ATAC sequencing

The ATAC libraries were generated following the Omni-ATAC protocol[103]. Transposase (Tn5) with assembled Illumina-compatible primers was produced by the Proteomics and Biochemistry Platform of the Andalusian Center for Developmental Biology (CABD, UPO/CSIC/JA) following a published protocol[104]. The WT, KO, and tKO hPSCs were dispersed to single cells using TrypLE Select (Thermo Fisher Scientific, Netherlands), counted, spun down at $500 \times g$ for 5 min at 4 °C and resuspended at a concentration of $5 \times 10^4$ cells/50 μl per biological replicate in cold ATAC-RSB: 10 mM Tris–HCl (BioShop, Canada) pH = 7.4, 10 mM NaCl (BioShop, Canada), 3 mM MgCl₂ (Millipore, Germany), containing 0.1% (v/v) Tween-20 (Roche Diagnostics, Germany), 0.1% (v/v) NP-40 (Biosciences, USA). For the PP stage, WT and KO cells were cryopreserved at day 12 of differentiation and thawed according to a previously published protocol[105]. For nuclei extraction, the cells were incubated with 0.1% (v/v) digitonin (Promega, USA) for 3 min at 4 °C and centrifuged at $500 \times g$ for 10 min at 4 °C with the addition of 1 ml ATAC-RSB, containing 0.1% (v/v) Tween-20. The supernatant was discarded, and nuclei were prepared for the transposition reaction as described previously[103]. The solution containing transposed fragments was purified using the Zymo DNA Clean & Concentrator (Zymo Research, USA), and the transposed DNA was eluted in 20 μl H₂O and stored at −20 °C. Barcoded adapters[103] were added to the transposed fragments by PCR with the NEBNext Ultra II Q5 2× Master Mix (New England Biolabs, USA), according to the producer protocol. All samples were amplified for 10 cycles, followed by a double-sided purification with the AMPure XP beads (Beckman Coulter, USA) and eluted in 20 μl H₂O. Concentration was established with Qubit dsDNA HS Assay kit (Thermo Fisher Scientific, Netherlands) and the quality of the samples was assessed using TapeStation HS D1000

kit (Agilent Technologies, USA). The samples were 150 bp paired-end sequenced on the NovaSeqX platform. The RNA-seq data have been deposited in the NCBI GEO database under accession number GSE227794. The nfcore/atacseq bioinformatic pipeline was used for the ATAC-seq data analysis[106]. Briefly, after adapter trimming with Trim Galore and quality check with FastQC, the reads were mapped to the human genome (hg38) using Bowtie2 with default paired-end settings. Next, all non-nuclear reads and unmapped paired reads were discarded. Duplicated reads were removed with picard MarkDuplicates. Filtering to remove unmapped regions, mitochondrial DNA, and blacklisted regions was done using SAMtools. Macs2 was used for broad peak calling and HOMER for peak annotation (annotate-Peaks.pl). Motif enrichment analysis was conducted with Homer (findMotifsGenome.pl). The normalized bigWig files, scaled to 1 million mapped reads, were created with BEDTools. Differential accessibility analysis was carried out using DESeq2. IGV platform was used for the visualization of bigWig tracks, peaks, and differential sites.

### Reverse transcription-quantitative PCR (qPCR)
Total RNA was isolated with Trizol reagent (Thermo Fisher Scientific, Netherlands) according to the manufacturer's instructions. For cDNA synthesis, 500 ng of RNA was reverse transcribed with SuperScript II Reverse Transcriptase cDNA synthesis kit (Thermo Fisher Scientific, Netherlands), with random hexamer primers. The Power SYBR Green qPCR Master Mix (Thermo Fisher Scientific, Netherlands) was used for cDNA amplification. The primer sequences are listed in Supplementary Table 5. The amount of target gene in each sample was normalized against a β-ACT control based on ΔCT quantification methods.

### ETV1 OE in hPSCs
*ETV1* cDNA was amplified from total HUES8 hESC cDNA with the following primers:

For transient plasmid-based *ETV1* overexpression:
hETV1_OE_FORWARD:
AGCGCTACCGGACTCAGATCCCACCATGGATGGATTTTATGACCAG
hETV1_OE_REVERSE:
CGTCGTCATCCTTGTAATCACCTCCATACACGTAGCCTTCGTTG
For *piggyBac* transposon-based *ETV1* overexpression:
hETV1_OE_FORWARD_1:
CTTTAAAGGAACCAATTCAGccaccATGGATGGATTTTATGAC
hETV1_OE_REVERSE_1:
GTACAAGAAAGCTGGGTCTAGA-
TATCTTActtatcgtcgtcatccttgtaatcacgATACACGTAGCCTTCGTTGT

The primers were designed with the Benchling Assembly Wizard tool. *ETV1* cDNA was amplified by PCR with the Q5 polymerase (NEB, USA) according to the manufacturer's instructions, with 2 steps PCR cycling conditions[107]. The cycling conditions were as follows: 98 °C for 30 s, followed by 15 cycles of 98 °C for 15 s, 63 °C for 20 s, 72 °C for 90 s, 25 cycles of 98 °C for 15 s, 72 °C for 15 s, 72 °C for 90 s, and the final extension step at 72 °C for 5 min. The resulting *ETV1* PCR product was isolated with the Monarch DNA Gel Extraction Kit (NEB, USA).

### Doxycycline-inducible ETV1 overexpression in hPSCs
Plasmid Gateway pENTR 1 A (Invitrogen, Netherlands) was digested with the *SalI* and *XhoI* restriction enzymes (Thermo Fisher Scientific, Netherlands). NEBuilder HiFi DNA Assembly Cloning Kit (NEB, USA) was used to ligate the *ETV1* PCR product with vector. Next, Gateway pENTR 1 A plasmid with *ETV1* cDNA sequence was used to LR recombination reaction with plasmid PB-TA-ERP2 (Addgene, USA, 179966)[60] resulting in the final plasmid (*ETV1*-OE plasmid) ensuring establishment of *ETV1* inducible overexpression in WT HUES8-iCas9 and *ETV1* KO hESCs (WT_OE and KO_OE, respectively). hESCs were transfected with the Lipofectamine STEM kit (Thermo Fisher Scientific, Netherlands) according to the manufacturer's protocol, using *ETV1*-OE plasmid and pCMV-hyPB plasmid[108] (gift from Dr. Allan Bradley) carrying

transposase enzyme. Next, puromycin treatment (1 μg/ml) was used to select positively transfected cells. To induce *ETV1* upregulation hPSCs were treated with 0.5 μg/ml doxycycline for a period of 24 h.

### Transient ETV1 overexpression
Plasmid pEGFP_N1_FLAG (gift from Dr. Patrick Calsou (Addgene, USA, 60360)[109]) was digested with the *XhoI* and *MluI* restriction enzymes (Thermo Fisher Scientific, Netherlands). NEBuilder HiFi DNA Assembly Cloning Kit (NEB, USA) was used to ligate *ETV1* PCR product with vector. hPSCs were transfected with the Lipofectamine STEM kit (Thermo Fisher Scientific, Netherlands) according to the manufacturer's protocol. To identify positively transfected single cells, GFP+ cells were analyzed in an IncuCyte live-cell imager (Sartorius, Germany) with a 2 h interval for taking images.

### Statistics
All the graphs were prepared in GraphPad Prism v8.4.2 for Windows (GraphPad Software, Boston, MA USA). For statistical analyses, unpaired two-tailed Student *t*-tests or one-way ANOVA for multiple comparisons were performed. All values are shown as means ± SDs.

### Reporting summary
Further information on research design is available in the Nature Portfolio Reporting Summary linked to this article.

## Data availability
Raw RNA-seq, scRNA-seq, and ATAC-seq data generated during the study have been deposited in the NCBI GEO database under accession number GSE227794. Raw image files are available from the corresponding author upon request. The authors declare that cells are available for the research community upon a request, from the corresponding author. Source data are provided with this paper.

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

## Acknowledgements

We thank Dr. Patrick Calsou (CNRS, France) for sharing the plasmid. We also thank Dr. Wojciech Szlachcic for valuable discussions, and Dr. Maciej Lalowski, and Dr. Jolanta Chmielowiec for critical manuscript reading. We would like to express our gratitude to Dr. Artur Jankowski for excellent technical support and all members of Borowiak lab for valuable discussions and support. This project was carried out within the TEAM program of the Foundation for Polish Science co-financed by the European Union under the European Regional Development Fund (POIR.04.04.00-00-20C5/16) to M.B. and Polish National Science Center (NCN) OPUS program 2019/33/B/NZ3/01226, 2020/37/B/NZ3/01917, all to M.B., National Science Center (Polonez UMO-2015/19/P/NZ3/03452) and UE Horizon 2020 (MSCA grant 665778), the Foundation for Polish Science - TEAM - program financed by the European Union within the European Regional Development Fund TEAM to M.B., and Minigrant financed by European Union under the "Passport to the future - Inter-disciplinary doctoral studies at the Faculty of Biology, Adam Mickiewicz University" (POWR.03.02.00-00-I006/17) and Polish National Science Center (NCN) Miniatura program 2023/07/X/NZ3/01135 for N.Z. Work in the laboratory of A.W. was supported by NSF grant MCB- 2135296 and NIH grant 5R35GM149328. Work in the laboratory of K.K.N. was supported by the Wellcome 221856/Z/20/Z (K.K.N.). For Open Access, the authors have applied a CC BY public copyright license to any Author Accepted Manuscript version arising from this submission.

## Author contributions

N.Z. – experimental design, performed experiments, data analysis, figure preparation, and manuscript writing, M.S. – performed ATAC-seq, scRNA-seq analysis, tKO pancreatic differentiation, data analysis, figure preparation, and manuscript writing, K.B. – performed RNA-seq experiments, RNA-seq and ATAC-seq data analysis, and figure preparation, D.K. – generated *ETV1/ETV4/ETV5* tKO hPSCs and performed flow cytometry analysis of ETV1 OE hPSCs, E.U. – performed the *ETV1* transient overexpression experiments, S.H. – performed spontaneous differentiation, M.G. – performed selected western blot analysis. A.W. – gastruloid experimental design, data analysis, manuscript editing, M.C.G. – performed gastruloid experiments. K.K.N – experimental design and data analysis, manuscript editing, M.B. – experimental design, data analysis, funding acquisition, and manuscript writing.

## Competing interests

The authors declare no competing interests.
