## [Transparent Peer Review file · Nature Communications]

ETVs dictate hPSC differentiation by tuning biophysical properties

Corresponding Author: Professor Malgorzata Borowiak

Version 0:

Reviewer comments:

Reviewer #1

(Remarks to the Author)

Review Nat Commun manuscript (Ziojła et al.)

ETV transcription factors exert various functions in normal physiological processes, including branching morphogenesis, embryonic development, neural system development, and cell metabolism. Numerous studies of ETV genes have been conducted in mice, but the role of ETVs in humans are relatively unknown. The manuscript from the Borowiak laboratory reports that ETV transcription factors in human pluripotent stem cells (hPSCs) control biophysical parameters via PI3K/AKT signaling and regulate multilineage differentiation. The authors demonstrate that genetic ablation of ETV1 or ETV1/ETV4/ETV5 in hPSCs enhances cell-adhesion, increases cellular tension, and impairs germ layer organization with ectoderm loss and extraembryonic overgrowth. Subsequently, ETV1-KO hPSCs progressively differentiated into definitive endoderm (DE) and pancreatic progenitors (PP) but not endocrine progenitors (EP) cells. While presenting several interesting phenotypes, this work, in my view, has some important issues with this study. The downstream analysis predicted from ETVs binding motif lacks in certainty and accuracy, making the results difficult to interpret and to evaluate whether this analysis really demonstrates the role of ETVs. The authors also fell short of providing sufficient explanations about how multilineage differentiation was impaired in ETVs-KO cells. Therefore, I would not recommend acceptance of this manuscript. Further work is needed to provide more reliable results for ETV function as well as a more detailed characterization of their downstream target genes. Please see below for further details.

Major comments

1. Title: "ETVs dictate hPSC fate by tuning biophysical properties" is not accurate. hPSC fate requires self-renewal and differentiation capacities. The authors could derive ETVs-KO cell lines, suggesting that ETVs-KO does not impair self-renewal.
2. Introduction section is quite redundant and unclear. The authors should provide a clear justification for why ETV transcription factors are needed for mechanobiology studies. In mouse studies, cell-adhesion and cellular tension seem not to be a major role of ETV transcription factors.
3. Could the authors rescue the ETVs-KO phenotypes using Doxycycline-inducible gene expression (Tet-On) system induced to ETVs-KO cell lines?
4. In Fig 3G, the gene classification of adhesion, pluripotency, and differentiation is unclear. The authors should clearly show whether ETVs-KO cells expressed pluripotency markers, which differentiated cell lineage markers they expressed, and which adhesion genes are essential for ETVs-KO phenotypes.
5. Although the authors computationally predicted downstream genes using ETVs binding motif, this method lacks in certainty and accuracy. The authors should conduct ChIP-seq of ETV1, ETV4, and ETV5 to identify downstream genes. Since ETVs are known as multifunctional genes in embryonic development, the authors should confirm the downstream genes in each differentiation stage using ChIP-seq and/or ChIP-qPCR.
6. Conducting an additional quantitative analysis for membrane and cytoskeletal proteins (e.g., ITGA5, VCL, and CDH1) would be more accurate using western blot.
7. In Fig 5A-C, how ETVs-KO impaired germ layer organization with ectoderm loss and extraembryonic overgrowth is not carefully characterized. Further analysis of gastruloids is needed.
8. In Fig 5 G-K, the authors derived pancreatic cells only from ETV1-KO hPSCs. ETV1/ETV4/ETV5-KO cell lines should be applied to this analysis.
9. In Fig 5 G-K, it will be important to study how endocrine progenitors (EP) differentiation was impaired in ETV1-KO cells.
10. While endocrine progenitors (EP) were not derived from ETV1-KO hPSCs in Fig 5, subsequent scRNA-seq identified

some amount of EP population (shown as EPandEC in Fig 6) in ETV1-KO cells. This discrepancy should be carefully discussed. It is also hard to believe that changes in protein levels of VCL and COL4A can lead to critical differences in pancreatic progenitor (PP) maturation. As far as observed from scRNA-seq data, there seems to be no significant phenotype in ETV1-KO cells.

Minor comments

1. Scale bar is missing in Fig 2A and Fig 5B.
2. The authors should avoid "data not shown" statements and make their data visible.

Reviewer #2

(Remarks to the Author)

The authors have identified ETV1/4/5 as being essential mediators of adhesion signalling within hPSCs. The triple knockout has a dramatic increase in cell adhesion, both cell-cell and cell-ECM adhesion, and shows disrupted differentiation capacity in 2D gastruloid models. There is interesting 3D models showing very different capacity for adhesion and multicellular organisation, and it is noted the cells are less amenable to enzymatic digestion. ETV1 overexpression shows poorer adhesion by contrast. There is also a noted deficiency in being able to make pancreatic progenitors from hPSCs with connection to signals such as Notch and PI3K/Akt.

This work is another solid piece in an emerging line of work exploring the role of ECM and mechanics in regulating hPSC fate and function. It will be of great interest to the field.

I feel the conclusions and claims are fairly well backed up, though I don't believe the authors have done enough on the mechanics side to support claims about 'cellular tension' such as they made in the abstract. It's not necessary anyway and the authors ought to remove such language. It is difficult to say what's happening here in terms of mechanoregulation of cell adhesions (adhesions could be affected in multiples ways), and the authors did little in the way of dynamics. Thus inferences about tension, migration, etc are not appropriate. The authors should keep their powder dry here and focus on adhesion specifically independent of tension. Their results likely imply major differences in cell mechanics but this is not shown (other than images of actin and vinculin, etc, which are not sufficient to make broad conclusions). The data on adhesion is good, and sufficient to support their conclusions. There could be a discussion about the implications about what this might mean for cell tension/mechanics.

The data analysis is good, they have performed a good analysis validating their perturbations, and the methodology is overall very good.

The paper doesn't present much insight into why ETV1/4/5 might be changed in physiological conditions, i.e. what is upstream. This is fairly well known anyway. I do not believe it to be necessary and they are clear they are presenting these factors as an essential hub between whatever extracellular cues impinge on the cells and intracellular signalling so I think it's fine to leave this for a subsequent study.

It's a very interesting phenotype presented here, with possibly important implications in outside-inside signalling. It's likely to be considered important by the field at large. I strongly support publication. I only ask the authors to soften any claims about mechanics specifically since it was not studied thoroughly here, and I don't think there is any justification for the authors to expand their work to provide claims that would justify those conclusions.

Reviewer #3

(Remarks to the Author)

The manuscript by Ziojła et al describes the role of ETV genes in modulating the biophysical properties of hPSCs, the three germ layers and hPSC-derived pancreatic cells. The authors use a robust approach by assessing the roles of ETV1 separately (KO) or in combination with ETV4 and ETV5 (tKO) and identify defects in cell adhesion and/or cell fate at the different stages analysed. Interestingly, the authors observed disruption in germ layer organisation in KO gastruloids, the complete absence of ectoderm in tKO gastruloids, impaired pancreatic cell differentiation, and provide mechanistic explanations including the dysregulation of various signalling pathways. The text is clear and the data is well-presented. One of the strongest aspects of the manuscript is that PI3K/AKT signalling is dysregulated upon ETV gene deletion and the mechanistic insight this provides. Overall, the paper provides a novel contribution to the field about the role of ETV genes during development.

Recommendations to strengthen the article, with a focus on the role of ETV genes in pancreatic cell differentiation:

1) Two separate protocols were used for the generation of endoderm, the first was in the form of gastruloids, and a separate protocol was used to form endoderm prior to pancreatic progenitor differentiation. A significant upregulation of endoderm markers was observed in the gastruloid protocol while no significant differences were observed between WT and KO cells in the pancreas cell differentiation protocol. To isolate a pancreas-specific role, the authors checked Sox17 levels by immunofluorescence staining and determined that DE induction was of a similar efficiency using the PP protocol for both WT and KO cells. Were any qPCRs done/other markers checked at this stage as was done in the gastruloid model, and was primitive gut tube induction efficiency similar between WT and KO cells? Could the previous observation that hPSCs are themselves affected by ETV1 deletion (e.g. impaired adhesion) have impacted the rest of the differentiation?

2) Did the authors use a HUES8-iCas9 control cell line that also underwent lipofection with a backbone plasmid minus the sgRNA for the PP differentiation? It would be useful to clarify this. Was the inducible feature of this cell line utilised at all for

the PP differentiation?

3) The requirement for ETV1 in the formation of endocrine progenitors may simply be a result of its requirement for pancreatic progenitor formation. The authors acknowledge this in the text and subsequently focus on pancreatic progenitors in which they identify a downregulation of PDX1 and NKX6.1 in the KO. Perhaps the abstract could be modified slightly to reflect this? A comparison of PDX1 and NKX6.1 expression levels between PP1 and PP2 on the dotplot in Fig 6C shows that NKX6.1 expression is much lower in PP1 than in PP2, while it was actually PP2 that was increased by 80% in the KO. Perhaps the authors could comment briefly on this (i.e. based on the flow cytometry plots, you would expect PP2 cells to express less NKX6.1 than the WT cells).

4) Similarly, would it be possible to provide a brief explanation as to why PP2 may be increased in the KO, and to perhaps comment on the role of increased NOTCH/ SLIT in pancreatic progenitor cells. For example, it has previously been shown that the activation of Notch signalling prevents differentiation into endocrine and exocrine cell lineages ([https://doi.org/10.1016/S0012-1606\(03\)00326-9](https://doi.org/10.1016/S0012-1606(03)00326-9)). High Notch activity is also known to inhibit Ngn3 transcription (<https://doi.org/10.1242/dev.078634>).

5) The fold change in cell number of the different cell populations identified using scRNA-seq is interesting (Fig. 6E), in particular the approximately 4-fold increase in AFP+ liver progenitor cells and the increase in mesenchyme. Perhaps a sentence or two in the discussion about the emergence of these cells in the KO model and how this result fits in with the overall role of ETV1 in cell adhesion would benefit the manuscript. Did the authors consider that the KO line may differentiate better into non-pancreatic lineages such as hepatocytes?

6) How do the authors discern whether the disruption to PI3K/AKT signalling at the PP stage is not an effect carried forward from the endoderm stage? This at the end of the introduction sounds as though the impairment to cell adhesion is different in the germ layers vs in hPSC-derived PPs, I suggest modifying the text to say it is enhanced in both:
"Specifically, ETV loss alters cytoskeleton, enhances cell-cell and cell-ECM adhesion via PI3K/AKT signaling and hinders proper differentiation into three germ layers. Moreover, ETV1 knockout (KO) impairs cell-cell and cell-ECM adhesion during in vitro pancreatic differentiation, markedly reducing EP formation."

Some minor points:

- Is there an explanation for why iPSC colonies in the tKO condition looked looser if cell adhesion markers increase upon ETV gene deletion, and why the same loose structure was observed upon ectopic ETV1 expression? Perhaps this can be explained a little in the text.

- The scale bars are missing in Figure 5B.

- May be useful to check the definite/indefinite articles the/a are present in the text, and the correct use of the plural e.g. "Our findings indicate that disrupting ETV1 or combination of ETV1/ETV4 /ETV5".

Version 1:

Reviewer comments:

Reviewer #1

(Remarks to the Author)

The authors have thoughtfully addressed all previous comments and suggestions. The paper is now clearer and stronger than the original version. I feel this manuscript is suitable for publication in Nature Communications now.

Reviewer #3

(Remarks to the Author)

Thank you for addressing each point. I'm very happy with the revised manuscript and think it is ready to be accepted for publication.

We thank the Reviewers for their time and valuable input on our manuscript. We are delighted that Reviewers: 2 and 3, find our work exciting and suitable for publication in Nature Communications. We appreciate the suggestions and hope Reviewer 1 will be satisfied with our additions and revisions. We have addressed all your comments; please find the schedule of our responses below, highlighted in blue.

REVIEWER COMMENTS

Reviewer #1 (Remarks to the Author):

Review Nat Commun manuscript (Ziojła et al.)

ETV transcription factors exert various functions in normal physiological processes, including branching morphogenesis, embryonic development, neural system development, and cell metabolism. Numerous studies of ETV genes have been conducted in mice, but the role of ETVs in humans are relatively unknown. The manuscript from the Borowiak laboratory reports that ETV transcription factors in human pluripotent stem cells (hPSCs) control biophysical parameters via PI3K/AKT signaling and regulate multilineage differentiation. The authors demonstrate that genetic ablation of ETV1 or ETV1/ETV4/ETV5 in hPSCs enhances cell-adhesion, increases cellular tension, and impairs germ layer organization with ectoderm loss and extraembryonic overgrowth. Subsequently, ETV1-KO hPSCs progressively differentiated into definitive endoderm (DE) and pancreatic progenitors (PP) but not endocrine progenitors (EP) cells. While presenting several interesting phenotypes, this work, in my view, has some important issues with this study. The downstream analysis predicted from ETVs binding motif lacks in certainty and accuracy, making the results difficult to interpret and to evaluate whether this analysis really demonstrates the role of ETVs. The authors also fell short of providing sufficient explanations about how multilineage differentiation was impaired in ETVs-KO cells. Therefore, I would not recommend acceptance of this manuscript. Further work is needed to provide more reliable results for ETV function as well as a more detailed characterization of their downstream target genes. Please see below for further details.

Major comments

1. Title: "ETVs dictate hPSC fate by tuning biophysical properties" is not accurate. hPSC fate requires self-renewal and differentiation capacities. The authors could derive ETVs-KO cell lines, suggesting that ETVs-KO does not impair self-renewal.

We initially used the term "fate" because ETV KO affects multiple aspects of hPSC behavior, including growth characteristics, cell-cell and cell-matrix interactions, and differentiation potential, particularly towards pancreatic and definitive endoderm lineages. While ETV KO lines can be derived and maintained, indicating that self-renewal is not abolished, the observed alterations in these key properties collectively influence the overall fate of hPSCs. To avoid any ambiguity, we have revised the title to "ETVs regulate hPSC differentiation by tuning biophysical properties" to more accurately reflect the focus of our study.

2. Introduction section is quite redundant and unclear. The authors should provide a clear justification for why ETV transcription factors are needed for mechanobiology studies. In mouse studies, cell-adhesion and cellular tension seem not to be a major role of ETV transcription factors.

Thank you for your thoughtful feedback. I appreciate the opportunity to clarify it. We have revised the Introduction and made it more concise. Regarding second part of you comment: ETV transcription factors have indeed been shown to regulate the pluripotency of mouse ESCs, and play roles in processes such as lung morphogenesis and cancer metastasis. As highlighted in the Introduction, our scRNA-seq data from the developing murine pancreas initially drew our attention to ETVs, as they were differentially expressed in specific subpopulations of endocrine progenitors undergoing tissue remodeling.

This finding prompted us to investigate ETV1, along with a few other candidate genes, in the context of mechanisms regulating mechanosignaling and remodeling. “[Redacted]” These observations suggest that ETV transcription factors have significant roles in modulating mechanobiological properties.

We believe the foundation for studying the role of ETV transcription factors in mechanobiology has been well-established in the Introduction, and our findings contribute to further elucidating their involvement in mechanosignaling and remodeling. Having said that, if there are specific areas where the Reviewer feels the justification still can be improved, we would be happy to address it.

3. Could the authors rescue the ETVs-KO phenotypes using Doxycycline-inducible gene expression (Tet-On) system induced to ETVs-KO cell lines?

We performed inducible overexpression (OE) of ETV1 in the WT and ETV1 KO background and observed at least partially restored the expression of CDH1 and ITGA5 in the OE-KO lines. Conversely, ETV1 OE in WT hPSCs decreased levels of these adhesion markers, suggesting that expression of CDH1 and ITGA5 is ETV1-dependent in hPSCs. These data are now presented in **Figure 2L** and **Supplementary Figure 2G**.

4. In Fig 3G, the gene classification of adhesion, pluripotency, and differentiation is unclear. The authors should clearly show whether ETVs-KO cells expressed pluripotency markers, which differentiated cell lineage markers they expressed, and which adhesion genes are essential for ETVs-KO phenotypes.

To address the Reviewer's concern regarding the clarity of data presentation in **Figure 3**, we have reorganized the datasets by grouping genes according to their functional roles and revised the figure (**new Figure 3D-F**). For enhanced clarity, we have also substituted bubble plots with heatmaps that highlight changes in the expression of genes involved in pluripotency, lineage commitment, and cell/ECM-cell adhesion in both the KO and tKO cells (**new Figure 3G-H**). These heatmaps show only significant differentially expressed genes for each comparison (KO

vs. WT and tKO vs. WT) and provide a clearer visualization of the magnitude and direction of gene expression changes compared to bubble plots. Moreover, the heatmaps indicate Gene Ontology (GO) terms for particular genes facilitating interpretation. Additionally, we have included a comprehensive supplementary table detailing the differentially expressed genes in KO and tKO, categorized by their affected GO terms (**Supplementary Table 1**).

5. Although the authors computationally predicted downstream genes using ETVs binding motif, this method lacks in certainty and accuracy. The authors should conduct ChIP-seq of ETV1, ETV4, and ETV5 to identify downstream genes. Since ETVs are known as multifunctional genes in embryonic development, the authors should confirm the downstream genes in each differentiation stage using ChIP-seq and/or ChIP-qPCR.

We appreciate your inquiry about the identification of ETV direct targets. We initially attempted to identify ETV1-bound regions using ChIP-seq. Despite rigorous efforts, including testing three different antibodies, optimizing various protocols, and getting successful controls, we consistently obtained a poor yield of the ETV1-bound fraction, hindering our ability to generate robust ChIP-seq data. This difficulty led us to employ ChIP-qPCR, as presented in the original submission (**Figure 3L**). To understand the limitations of the ETV-specific ChIP-seq approach, we investigated the cellular distribution of ETV1 protein. A recent study revealed that ETS1 protein, and ETVs belong to the ETS family, is present in both chromatin-bound and soluble fractions, with a substantial proportion residing in the unbound state¹. Our own analysis of hPSCs confirmed this observation, showing even higher levels of ETV1 in the soluble fraction compared to chromatin fraction (**revised Figure 3K**). This finding might explain the challenges in obtaining high-quality ChIP-seq data for ETV1 in hPSCs.

While ChIP-seq data are not currently available, we have included alternative evidence to support our conclusions. Specifically, we provide ATAC-seq data for both hPSCs and PPs, demonstrating differential chromatin accessibility at key regulatory regions (**revised Figure 3I-J** for hPSCs, **revised Figure 6I**, and **Supplementary Figure 6D** for PPs). These data complement our ChIP-qPCR findings and provide further support for the role of ETV in regulating adhesion-related gene expression during these developmental stages.

6. Conducting an additional quantitative analysis for membrane and cytoskeletal proteins (e.g., ITGA5, VCL, and CDH1) would be more accurate using western blot.

As requested, we performed western blot analysis of VCL (**revised Figure 4C** and **Supplementary Figure 4C**) and CDH1 (**Rebuttal Figure 1**), and quantification of these signals confirmed that VCL and CDH1 protein levels were increased in KO and tKO hPSCs.

Rebuttal Figure 1. Western blot analysis of CDH1 protein expression in WT, KO, and tKO hPSCs. GAPDH serves as a loading control. Quantification of CDH1 protein levels normalized to GAPDH is shown on the right.

7. In Fig 5A-C, how EVT_s-KO impaired germ layer organization with ectoderm loss and extraembryonic overgrowth is not carefully characterized. Further analysis of gastruloids is needed.

In collaboration with Dr. Warmflash we carefully phenotyped gastruloids. This analysis pointed to aberrant formation of all three germ layers (**new Figure 5A-C** and **Supplementary Figure 5A**). We believe data included in the MS are sufficient to appreciate the effect of the ETVs KO on gastruloid organization and substantiate the causative link between *ETVs* and the observed developmental changes.

Concerning additional experiments, we tested two major pathways, i.e. BMP signaling and WNT pathway in the gastruloid model. We have not seen a difference in pSmad1 expression, however a high upregulation of LEF1 was observed 6h after BMP4 treatment in tKO, while the single KO showed a mild upregulation compared to WT gastruloids (**Rebuttal Figure 2**).

Rebuttal Figure 2. Quantification represents intensity levels of LEF1 normalized to DAPI, averaged at different positions along the colony radii. Error bars represent standard error of the mean. Samples were immunostained for the indicated marker at 2, 6 and 24 h post BMP treatment. A high and mild upregulation was observed in tKO and the single KO, respectively, compared to WT gastruloids.

Interestingly, WNT was one of the dysregulated pathways in hPSC RNA-seq analysis. The Wnt signaling pathway regulates cell adhesion in part through its effect on β -catenin, a protein that plays a dual role in both signaling and cell-cell adhesion. In the presence of Wnt signals, β -catenin accumulates in the cell and translocates to the nucleus to drive gene expression; however, in the absence of Wnt, β -catenin is degraded and instead localizes to cell-cell

junctions where it binds cadherins to stabilize adherens junctions, supporting cell adhesion². Analysis performed by means of the Ingenuity Pathway Analysis software (IPA, Qiagen) predicted upregulation in the activity of WNT/ β -catenin pathway in KO compared to WT. The tKO vs. WT comparison predicted mixed response, with activation and inhibition at different levels of the WNT/ β -catenin signaling (**Rebuttal Figure 3**). This goes in line with all the data we have included in the MS, that identify altered adhesion patterns in all the ETV KO cell lines.

A

B

C

Rebuttal Figure 3. RNA-seq analyses of ETV1 KO and tKO using IPA software. A) A predicted upregulation in the activity of WNT/β-catenin pathway is observed in ETV1 KO (z-score >2). B) Mixed response in the pathway activity is observed in the tKO cells (middle panel). The prediction is based on measured logFC changes in the transcript expression vs. WT, at p-value adj.<0.05 (shown by the selected network nodes). C) Network nodes and relationships legend.

8. In Fig 5 G-K, the authors derived pancreatic cells only from ETV1-KO hPSCs. ETV1/ETV4/ETV5-KO cell lines should be applied to this analysis.

Thank you for this suggestion. While we initially believed the original submission contained sufficient data, we understand the importance of further elucidating the pancreatic differentiation aspect. Therefore, we have now included data on the pancreatic differentiation of tKO hPSCs, as requested. These findings reveal a severe impairment in pancreatic cell differentiation in tKO, characterized by a scarcity of NKX6.1+, CHGA+, and INS+ cells. Specifically, we observed a significant decrease in the expression of SOX17, a definitive endoderm marker, in tKO cells differentiated using two protocols and two cell culture formats: 2D (**revised Figure 5D**) and 3D (**Rebuttal Figure 3**). The defective endoderm induction might impact the pancreatic differentiation. Moreover, we observed significant morphological alterations in the differentiating spheroids, including irregular shapes and the frequent formation of hollow spaces. Interestingly, in contrast to the consistent phenotype observed in ETV1 KO, the tKO pancreatic differentiation showed some variability. The majority of the tKO spheroids exhibited severe differentiation or morphological defects, while few appeared similar to WT spheroids. This variability might be attributed to complex changes in gene expression profiles and genetic interactions of ETVs. However, across all tKO spheroids, the number of CHGA+ cells remained markedly low. Due to the limited space, we now have included the tKO pancreatic differentiation data in the Rebuttal letter as **Rebuttal Figure 4**.

Rebuttal Figure 4. The tKO hPSC differentiation toward β -cells. The immunofluorescence staining and mean intensity quantification (N =3) of stage specific markers is shown. Hollow spaces in the tKO spheroids are marked by a yellow asterisk.

9. In Fig 5 G-K, it will be important to study how endocrine progenitors (EP) differentiation was impaired in ETV1-KO cells.

While a comprehensive understanding of the molecular mechanisms underlying defective pancreatic differentiation in ETV1 KO cells warrants a dedicated study (as also noted by Reviewer 3), and this is indeed an area of ongoing investigation in our lab. In the meantime, we have gathered for rebuttal, data suggesting that alterations in cell-cell and cell-ECM adhesion molecules such as VCL and COL4A disrupt cell-cell interactions and ECM organization, ultimately affecting morphogenesis and pancreatic progenitor differentiation. This is evidently manifested in the morphological differences observed between KO and WT pancreatic spheroids (**Rebuttal Figure 5**). Further, other studies have demonstrated that disruptions in adhesion or cell-cell contact affects murine pancreatic development^{3,4}.

The changes in adhesion related molecules might lead to changes in broad mechanosignaling network. In accordance with this, in ETV1 KO we observed changes in mechanosensitive pathways, such as PI3K/AKT and Hippo signaling (**Rebuttal Figure 6**). The data regarding PI3K are presented in the manuscript as part of scRNA-seq data (**Figure 6F**), in the analysis of bulk RNA-seq (**Figure 3D and G**), and an experimental follow-up with PI3K/AKT pathway agonist and antagonists (**Figure 4H-O**).

While the findings on Hippo signaling and overall morphogenetic changes of KO spheroids vs WT are intriguing, however, we have chosen not to include these data in the revised version of the manuscript. We believe that the manuscript already contains a substantial amount of data, and adding further details specifically on pancreatic differentiation could detract it from the focus of the study. We plan to present a more comprehensive analysis of the mechanisms underlying ETV1 role in pancreatic differentiation in follow-up publication.

Rebuttal Figure 5. Morphogenetic changes during *in vitro* differentiation of WT and ETV1-KO hPSCs into β cells.

“[Redacted]”

10. Progenitors (EP) were not derived from ETV1-KO hPSCs in Fig 5, subsequent scRNA-seq identified some amount of EP population (shown as EP_EC in Fig 6) in ETV1-KO cells. This discrepancy should be carefully discussed. It is also hard to believe that changes in protein levels of VCL and COL4A can lead to critical differences in pancreatic progenitor (PP) maturation. As far as observed from scRNA-seq data, there seems to be no significant phenotype in ETV1-KO cells.

Thank you for pointing out that it was not sufficiently discussed. In our scRNA-seq, we did observe expression of transcripts associated with endocrine fate. However, their expression was generally low, and importantly, by immunofluorescence we consistently detected very low to negligible expression of key endocrine proteins, including NKX6-1 and CHGA in KO cells (**Figure 5K-L** and **Supplementary Figure 5G**). Supporting these observations, we noticed differences in chromatin accessibility at the PDX1 and NKX6.1 *loci*, which are crucial for

pancreatic endocrine development (**Supplementary Figure 6D**). Furthermore, we found decreased levels of PDX1 protein per KO cell compared to WT (as measured by immunofluorescence and flow cytometry; **Figure 5K-L** and **Supplementary Figure 5G**). Together, these results suggest that despite the initial upregulation of a number of endocrine-related transcripts, the differentiation process is not progressing properly, and functional endocrine cells are not maturing and being formed.

Regarding second part of your comment, please note, that we partially addressed the impact of adhesion on pancreatic differentiation in our response, in point 9 of the rebuttal. Furthermore, it has been shown by Mamidi *et al.*⁵ that EP cell fate specification is regulated by mechanotransduction signaling pathways such as Hippo and Notch, among others. Depending on the ECM components, i.e. collagens, laminins, fibronectin, and vitronectin, the receptors present on the cell membrane and intracellular downstream signaling, PPs can either acquire the EP fate, multipotent progenitor fate or become ductal progenitors. In the case of Hippo signaling, which governs organ size and has a critical role in stem cell and tissue-specific self-renewal of progenitor cells, it has been demonstrated that its activity is dependent on cell adhesion, cell tension and cell-cell contacts. The major Hippo effector YAP1 in phosphorylated state is sequestered in the cytoplasm resulting in the Notch signaling inhibition, Ngn3 expression and EP fate activation⁶. On the contrary, non-phosphorylated YAP1 enters the nucleus, binds to one of its interacting partners (i.e. TEAD1), which enables higher Notch signaling activity. This subsequently hinders Ngn3 expression, critical for correct EP formation⁷. “[Redacted]”

Jeff Millman laboratory⁸ and others, demonstrated that cytoskeleton impacts the endocrine differentiation *in vitro*. Finally, there is a sizeable literature on how adhesion, cell-cell and cell-ECM communication regulate cell fate acquisition and development of different organs. “[Redacted]”

Minor comments

1. Scale bar is missing in Fig 2A and Fig 5B.

We apologize for these omissions, it has been amended.

2. The authors should avoid "data not shown" statements and make their data visible.

It has been fixed. Thank you for the comment.

Reviewer #2 (Remarks to the Author):

The authors have identified ETV1/4/5 as being essential mediators of adhesion signaling within hPSCs. The triple knockout has a dramatic increase in cell adhesion, both cell-cell and cell-ECM adhesion, and shows disrupted differentiation capacity in 2D gastruloid models. There is interesting 3D models showing very different capacity for adhesion and multicellular organisation, and it is noted the cells are less amenable to enzymatic digestion. ETV1 overexpression shows poorer adhesion by contrast. There is also a noted deficiency in being able to make pancreatic progenitors from hPSCs with connection to signals such as Notch and PI3K/Akt.

This work is another solid piece in an emerging line of work exploring the role of ECM and mechanics in regulating hPSC fate and function. It will be of great interest to the field.

I feel the conclusions and claims are fairly well backed up, though I don't believe the authors have done enough on the mechanics side to support claims about 'cellular tension' such as they made in the abstract. It's not necessary anyway and the authors ought to remove such language. It is difficult to say what's happening here in terms of mechanoregulation of cell adhesions (adhesions could be affected in multiples ways), and the authors did little in the way of dynamics. Thus inferences about tension, migration, etc are not appropriate. The authors should keep their powder dry here and focus on adhesion specifically independent of tension. Their

results likely imply major differences in cell mechanics but this is not shown (other than images of actin and vinculin, etc, which are not sufficient to make broad conclusions). The data on adhesion is good, and sufficient to support their conclusions. There could be a discussion about the implications about what this might mean for cell tension/mechanics.

The data analysis is good, they have performed a good analysis validating their perturbations, and the methodology is overall very good.

The paper doesn't present much insight into why ETV1/4/5 might be changed in physiological conditions, i.e. what is upstream. This is fairly well known anyway. I do not believe it to be necessary and they are clear they are presenting these factors as an essential hub between whatever extracellular cues impinge on the cells and intracellular signaling so I think it's fine to leave this for a subsequent study.

It's a very interesting phenotype presented here, with possibly important implications in outside-inside signaling. It's likely to be considered important by the field at large. I strongly support publication. I only ask the authors to soften any claims about mechanics specifically since it was not studied thoroughly here, and I don't think there is any justification for the authors to expand their work to provide claims that would justify those conclusions.

We thank the Reviewer for their thoughtful and encouraging evaluation of our manuscript. We are particularly grateful for their acknowledgment of the interesting phenotype we have presented and its potential implications for the field of hPSC biology and mechanosignaling. The impact of mechanosignaling and adhesion is still understudied in the field of hPSC differentiation. We agree with the Reviewer that our current data do not fully support the claims about "cellular tension" made in the abstract. We have carefully revised the manuscript and removed these statements to avoid overinterpretation of our findings. While our results suggest potential differences in cell mechanics, we acknowledge that further investigations are necessary to directly assess these properties. We have added a paragraph to Discussion section, to specifically address this point and discuss the potential implications of our findings for cell tension and mechanics.

We appreciate the Reviewer's feedback on the lack of investigation into the upstream regulators of ETV1/4/5. As the Reviewer notes, our focus was on establishing the role of these factors as a critical hub in adhesion signaling. We agree that a deeper exploration of upstream regulation is best suited for a subsequent study. We are grateful for the Reviewer's strong support for publication and believe the implemented revisions have further strengthened the manuscript.

New paragraph in the Discussion now reads: "It would be interesting to investigate cell tension in ETV1 KO or tKO hPSCs or PPs, as the increased adhesion likely affects cellular tension, further attuning the mechanosignaling pathways".

Reviewer #3 (Remarks to the Author):

The manuscript by Ziojła et al. describes the role of ETV genes in modulating the biophysical properties of hPSCs, the three germ layers and hPSC-derived pancreatic cells. The authors use a robust approach by assessing the roles of ETV1 separately (KO) or in combination with ETV4 and ETV5 (tKO) and identify defects in cell adhesion and/or cell fate at the different stages analysed. Interestingly, the authors observed disruption in germ layer organisation in KO gastruloids, the complete absence of ectoderm in tKO gastruloids, impaired pancreatic cell differentiation, and provide mechanistic explanations including the dysregulation of various signaling pathways. The text is clear and the data is well-presented. One of the strongest aspects of the manuscript is that PI3K/AKT signaling is dysregulated upon ETV gene deletion and the mechanistic insight this provides. Overall, the paper provides a novel contribution to the field about the role of ETV genes during development. Recommendations to strengthen the article, with a focus on the role of ETV genes in pancreatic cell differentiation:

1) Two separate protocols were used for the generation of endoderm, the first was in the form of gastruloids, and a separate protocol was used to form endoderm prior to pancreatic progenitor differentiation. A significant upregulation of endoderm markers was observed in the gastruloid protocol while no significant differences were observed between WT and KO cells in the pancreas cell differentiation protocol. To isolate a pancreas-specific role, the authors checked Sox17 levels by immunofluorescence staining and determined that DE induction was of a similar efficiency using the PP protocol for both WT and KO cells. Were any qPCRs done/other markers checked at this stage as was done in the gastruloid model, and was primitive gut tube induction efficiency similar between WT and KO cells? Could the previous observation that hPSCs are themselves affected by ETV1 deletion (e.g. impaired adhesion) have impacted the rest of the differentiation?

Thank you for bringing up these interesting aspects. We tested definitive endoderm formation in the absence of ETV1 and ETV1/ETV4/ETV5 using four different approaches, and in the initial submission, we included two of them. These approaches were as follows:

- Directed differentiation using a combination of Activin A and a Wnt agonist (“full protocol”), which also is a part of pancreatic differentiation protocol as the Reviewer noted,
- Gastruloids,
- Suboptimal directed differentiation using 1/3 the concentration of Activin A and no Wnt activation,
- Spontaneous differentiation based on embryoid bodies.

While we observed a clear difference using the gastruloid model, we did not see a major effect of ETV1 deletion during directed differentiation using the “full protocol”. We hypothesized that the high concentration of growth factors used during directed differentiation allowed cells to overcome some of the impairments or masked the potential differences between KO and WT cells. To test it, we performed spontaneous differentiation without any pro-endoderm growth factors added to the differentiation media and observed increased endoderm formation (based

on qPCR analysis of *SOX17*, *FOXA2*, and *GATA4*). Moreover, we performed directed differentiation using a suboptimal protocol with 1/3 the concentration of Activin A and no WNT agonist, and again we observed upregulation of SOX17+ cells. During gastruloid formation we did not add any Activin A and WNT agonist, only BMP4, which led to increased endoderm formation in ETV1 KO. We have added the new data on endoderm induction with suboptimal endoderm differentiation protocol to the MS: **new Figure 5D** (standard protocol), **new Figure 5E** (suboptimal protocol) and **new Supplementary Figure 5B** (suboptimal endoderm differentiation).

ETV1, by regulating hPSC mechanics/adhesion, impacts endoderm differentiation. Application of the full differentiation protocol for PP generation overrides any potential endoderm defect. In line with this, we have not detected any significant differences in the efficiency or timing of DE formation for ETV1 KO using the PP protocol. Please note that at day 8 of differentiation, which corresponds to the initiation of PDX1+ PP cell formation, we have not observed any measurable differences between ETV1 KO and WT cells. Together, these findings highlight the distinct roles of ETV1 in hPSCs and PPs.

2) Did the authors use a HUES8-iCas9 control cell line that also underwent lipofection with a backbone plasmid minus the sgRNA for the PP differentiation? It would be useful to clarify this. Was the inducible feature of this cell line utilised at all for the PP differentiation?

To clarify it, inducible Cas9 expression was utilized only during targeting of ETV genes at the ESC stage and not during pancreatic differentiation. Targeted clones with desired deletions were identified, selected, and expanded to obtain stable, clonal ETV KO hPSC lines, with permanent deletion of the target genes. These lines were subsequently passaged several times and used for differentiation experiments. To exclude off-target effects, we used two independent sgRNA sets targeting different regions of the ETV genes. Furthermore, HUES8-iCas9 hPSC line has been routinely used by other labs for gene editing studies^{9, 10}, demonstrating its reliability. Finally, to further validate our findings and to rule out the potential off-target effects specific to HUES8 line, we generated an ETV1 KO in H1 hESCs and observed the same defects.

3) The requirement for ETV1 in the formation of endocrine progenitors may simply be a result of its requirement for pancreatic progenitor formation. The authors acknowledge this in the text and subsequently focus on pancreatic progenitors in which they identify a downregulation of PDX1 and NKX6.1 in the KO. Perhaps the abstract could be modified slightly to reflect this? A comparison of PDX1 and NKX6.1 expression levels between PP1 and PP2 on the dotplot in Fig 6C shows that NKX6.1 expression is much lower in PP1 than in PP2, while it was actually PP2 that was increased by 80% in the KO. Perhaps the authors could comment briefly on this (i.e. based on the flow cytometry plots, you would expect PP2 cells to express less NKX6.1 than the WT cells).

Thank you for bringing up this interesting question. Similar point was also brought by the Reviewer 1, major comment #10. We are currently investigating the mechanisms of defective

PP maturation into EPs in the absence of ETV1. “[Redacted]” Overall, as we see comparable (in KO and WT) upregulation of PDX1 expression at day 8 of differentiation, we believe that the PPs are initially formed but their maturation into EPs is impaired in ETV1 KO. This disruption might also manifest as KO cells being stuck at PP2 stage, and accordingly, we observed a lower number of cells in the EP_EC cluster (based on scRNA-seq) and negligible *NGN3* expression at the protein level.

We also highlighted it in the Discussion: “Our scRNA-seq analysis of WT and ETV1 KO PPs suggest that ETV1 deficient PPs either exhibit arrested development at the PP2 stage, indicative of blocked endocrine lineage progression, or differentiate toward other alternative lineages, as evidenced by the acquisition of *CPA2* or *AFP* expression”.

4) Similarly, would it be possible to provide a brief explanation as to why PP2 may be increased in the KO, and to perhaps comment on the role of increased NOTCH/ SLIT in pancreatic progenitor cells. For example, it has previously been shown that the activation of Notch signaling prevents differentiation into endocrine and exocrine cell lineages ([https://doi.org/10.1016/S0012-1606\(03\)00326-9](https://doi.org/10.1016/S0012-1606(03)00326-9)). High Notch activity is also known to inhibit *Ngn3* transcription (<https://doi.org/10.1242/dev.078634>).

Please see our comment above.

5) The fold change in cell number of the different cell populations identified using scRNA-seq is interesting (Fig. 6E), in particular the approximately 4-fold increase in AFP+ liver progenitor cells and the increase in mesenchyme. Perhaps a sentence or two in the discussion about the emergence of these cells in the KO model and how this result fits in with the overall role of ETV1 in cell adhesion would benefit the manuscript. Did the authors consider that the KO line may differentiate better into non-pancreatic lineages such as hepatocytes?

We agree that the increased formation of hepatocyte-like and mesenchyme-like cells in the ETV1 KO, despite the presence of pancreatic differentiation signals, is an intriguing observation. We have also performed immunofluorescence staining for AFP at day 10 and observed the increased number of AFP+ cells in ETV1 KO condition (**Rebuttal Figure 8**). In separate line of investigations, we noted that alternations in ECM composition could impact pancreatic vs. hepatic cell fate during *in vitro* hPSC differentiation. We have now expanded the discussion in the manuscript to specifically address it, and plan to investigate this phenomenon in the future studies. We added to the Discussion: “Interestingly, scRNA-seq analysis also revealed the increase in hepatocyte-like and mesenchyme-like cells in the ETV1 KO cultures, even under conditions inducing pancreatic specification”.

Rebuttal Figure 8. Immunofluorescence staining for AFP (in green) at day 10 of ETV1 KO and WT hPSC pancreatic differentiation. Nuclei were stained with DAPI, in blue.

6) How do the authors discern whether the disruption to PI3K/AKT signaling at the PP stage is not an effect carried forward from the endoderm stage? This at the end of the introduction sounds as though the impairment to cell adhesion is different in the germ layers vs in hPSC-derived PPs, I suggest modifying the text to say it is enhanced in both: “Specifically, ETV loss alters cytoskeleton, enhances cell-cell and cell-ECM adhesion via PI3K/AKT signaling and hinders proper differentiation into three germ layers. Moreover, ETV1 knockout (KO) impairs cell-cell and cell-ECM adhesion during in vitro pancreatic differentiation, markedly reducing EP formation.”

Thank you for this comment. We have modified the Introduction as suggested and have also addressed this in our response to the Reviewer point 1.

Some minor points:

- Is there an explanation for why iPSC colonies in the tKO condition looked looser if cell adhesion markers increase upon ETV gene deletion, and why the same loose structure was observed upon ectopic ETV1 expression? Perhaps this can be explained a little in the text.

The Reviewer is correct to point out the seemingly contradictory changes in adhesion marker expression in the tKO. There, we observed dysregulation in the expression of adhesion markers, with many markers upregulated and some downregulated, collectively resulting in altered cell adhesion (**revised Figure 3F and H**). Please also note that the tKO is characterized by disrupted colony morphology with the appearance of "holes" (**Figure 4F** and quantification in **Supplementary Figure 4H**). The appearance of "holes" within tKO colonies further supports the idea that overall cell adhesion is compromised, despite the upregulation of some adhesion molecules. This complex phenotype highlights the intricate interplay between different adhesion molecules and the crucial role of ETV1, ETV4, and ETV5 in regulating the balance.

The revised MS now reads “The morphology of tKO colony was disrupted, with a higher presence of gaps (**Fig. 4F** and **Supplementary Fig. 4H**).”

The scale bars are missing in Figure 5B.

Thank you for letting us know about this omission. It has been fixed.

- May be useful to check the definite/indefinite articles the/a are present in the text, and the correct use of the plural e.g. "Our finding indicate that disrupting ETV1 or combination of ETV1/ETV4 /ETV5".

Thank you for pointing it out. We reviewed the manuscript carefully and corrected as needed.

References:

1. Li, X. et al. Proteomic analyses reveal distinct chromatin-associated and soluble transcription factor complexes. *Mol Syst Biol* **11**, 775 (2015).
2. Liu, J. et al. Wnt/beta-catenin signaling: function, biological mechanisms, and therapeutic opportunities. *Signal Transduct Target Ther* **7**, 3 (2022).
3. Tixi, W. et al. Coordination between ECM and cell-cell adhesion regulates the development of islet aggregation, architecture, and functional maturation. *Elife* **12** (2023).
4. Li, H., Neelankal John, A., Nagatake, T., Hamazaki, Y. & Jiang, F.X. Claudin 4 in pancreatic beta cells is involved in regulating the functional state of adult islets. *FEBS Open Bio* **10**, 28-40 (2020).
5. Mamidi, A. et al. Mechanosignaling via integrins directs fate decisions of pancreatic progenitors. *Nature* **564**, 114-118 (2018).
6. Hald, J. et al. Activated Notch1 prevents differentiation of pancreatic acinar cells and attenuate endocrine development. *Dev Biol* **260**, 426-437 (2003).
7. Shih, H.P. et al. A Notch-dependent molecular circuitry initiates pancreatic endocrine and ductal cell differentiation. *Development* **139**, 2488-2499 (2012).
8. Hoglebe, N.J., Augsornworawat, P., Maxwell, K.G., Velazco-Cruz, L. & Millman, J.R. Targeting the cytoskeleton to direct pancreatic differentiation of human pluripotent stem cells. *Nat Biotechnol* **38**, 460-470 (2020).
9. Li, Q.V. et al. Genome-scale screens identify JNK-JUN signaling as a barrier for pluripotency exit and endoderm differentiation. *Nat Genet* **51**, 999-1010 (2019).
10. Shi, Z.D. et al. Genome Editing in hPSCs Reveals GATA6 Haploinsufficiency and a Genetic Interaction with GATA4 in Human Pancreatic Development. *Cell Stem Cell* **20**, 675-688 e676 (2017).